# STAT1 regulates immune-mediated intestinal stem cell proliferation and epithelial regeneration

Shuichiro Takashima [1,2,11], Roshan Sharma [3,11], Winston Chang[1,4,11], Marco Calafiore[1,11], Ya-Yuan Fu [1,11], Suze A. Jansen [5,6], Takahiro Ito [1], Anastasiya Egorova[1], Jason Kuttiyara[1], Viktor Arnhold[1], Jessica Sharrock[4], Endi Santosa[4], Ojasvi Chaudhary [7], Heather Geiger[3], Hiromi Iwasaki[2], Chen Liu[8], Joseph Sun [4], Nicolas Robine [3], Linas Mazutis [7,9], Caroline A. Lindemans[5,6] & Alan M. Hanash [1,4,10] ✉

The role of the immune system in regulating tissue stem cells remains poorly understood, as does the relationship between immune-mediated tissue damage and regeneration. Graft vs. host disease (GVHD) occurring after allo-geneic bone marrow transplantation (allo-BMT) involves immune-mediated damage to the intestinal epithelium and its stem cell compartment. To assess impacts of T-cell-driven injury on distinct epithelial constituents, we have performed single cell RNA sequencing on intestinal crypts following experimental BMT. Intestinal stem cells (ISCs) from GVHD mice have exhibited global transcriptomic changes associated with a substantial Interferon-γ response and upregulation of STAT1. To determine its role in crypt function, STAT1 has been deleted within murine intestinal epithelium. Following allo-BMT, STAT1 deficiency has resulted in reduced epithelial proliferation and impaired ISC recovery. Similarly, epithelial Interferon-γ receptor deletion has also atte-nuated proliferation and ISC recovery post-transplant. Investigating the mechanistic basis underlying this epithelial response, ISC STAT1 expression in GVHD has been found to correlate with upregulation of ISC c-Myc. Further-more, activated T cells have stimulated Interferon-γ-dependent epithelial regeneration in co-cultured organoids, and Interferon-γ has directly induced STAT1-dependent c-Myc expression and ISC proliferation. These findings illustrate immunologic regulation of a core tissue stem cell program after damage and support a role for Interferon-γ as a direct contributor to epithelial regeneration.

Intestinal stem cells (ISCs), typically described as crypt base columnar cells expressing *Lgr5* or *Olfm4*, are critical for maintenance of the epithelial lining of the intestines[1–3]. They are accompanied within intestinal crypts by several other epithelial populations, such as secretory lineage cells possessing a range of functions including antimicrobial host defense and enterocyte lineage cells responsible for the absorptive functions of the surface epithelium[4]. Additionally, the secretory Paneth cell population is thought to provide an epithelial contribution to the stem cell niche by producing factors that impact ISC maintenance and function[5]. The ISC compartment is constitutively

active, continuously proliferating to maintain the intestinal lining, even during homeostasis[6,7].

Several mechanisms have been put forth to explain the robust capacity of the intestinal epithelium to regenerate after injury[7,8], but little is known about the mechanisms governing regeneration in response to immune-mediated damage. Notably, in addition to inducing damage, immune responses can also contribute to the epithelial regeneration that occurs following tissue damage[9–11]. It has been proposed that helper T cell (Th) responses may have distinct contributions to this process, with Type 2 (Th2) responses being important drivers of regeneration after damage and infection, Type 3 (Th17/22) responses contributing to regeneration but also being important for limiting the extent of injury that may develop, and Type 1 (Th1) responses primarily contributing to antimicrobial immunity while also inflicting some of the tissue damage that may then be repaired by the other branches of the immune response[12–16].

Allogeneic bone marrow transplantation (allo-BMT) is a potentially curative therapy for benign and malignant hematologic diseases as well as genetic disorders and even autoimmunity. However, this therapeutic option is limited by the risks of transplant-related toxicities such as graft vs. host disease (GVHD). In particular, acute gastrointestinal (GI) GVHD, which results from the donor immune system inflicting injury upon the transplant recipient's GI tract, is the predominant contributor to GVHD-related morbidity and mortality[17,18]. Crypt apoptosis and regeneration are both characteristic findings of GI GVHD[18], but the mechanisms driving this regeneration are poorly understood. ISCs and Paneth cells are both reduced in mice with GVHD[19–22], and Interferon-γ (IFNγ) is an important driver of this stem cell compartment injury, inducing apoptosis of both the stem cells and Paneth cells[23–25]. The pathways driving crypt regeneration in GVHD are thus unclear, and it is not known if this regeneration reflects merely a compensatory reaction to the immune-mediated damage or if the pathologic immune response may also be a direct contributor to the epithelial proliferation that occurs. The purpose of this study was to characterize the behavior of the ISC compartment and neighboring epithelial crypt components in response to immune-mediated injury and to investigate the contributions of tissue-targeted immunity to epithelial regeneration in the GI tract.

## Results

### Intestinal stem cell transcriptomes are profoundly impacted by immune-mediated GI damage

In order to investigate the response of the ISC compartment to immune-mediated injury within the context of other more differentiated epithelial components, we analyzed the transcriptomes of C57BL/6 (B6) crypt epithelial cells at baseline and after BMT using single cell RNA sequencing (scRNA-seq). Total body irradiation (TBI) was administered as pre-transplant conditioning, and then allo-BMT was performed utilizing marrow and T cells from MHC-mismatched donors (B10.BR) to induce GVHD in the GI tract. Transplants were also performed using syngeneic (syn) B6 donors to generate GVHD-negative controls receiving the same pre-transplant conditioning without the subsequent alloreactive immune response. Live single cells were isolated from small intestine (SI) crypts five days after BMT to investigate early events post-transplant that may drive the subsequent pathologic and regenerative manifestations. Crypt cells were also isolated from normal B6 mice during homeostasis to serve as baseline controls. The cells were then partitioned and barcoded for scRNA-seq using the 10x Genomics Chromium platform and sequenced (Fig. 1a). Following basic pre-processing of the data, the sequenced cells were visualized using Uniform Manifold Approximation and Projection (UMAP)[26] and clustered using PhenoGraph[27] (see "Methods") (Fig. 1b and Supplementary Fig. 1a; baseline shown in blue, syn-BMT in green, and allo-BMT in orange). Twenty-two clusters were identified and annotated based on published gene expression data sets

(Supplementary Fig. 1b)[28–31], thus identifying populations of ISCs, progenitors, mature enterocytes, secretory cells, and immune cells (Fig. 1b, c). Paneth cells and goblet cells initially clustered together but could be distinguished from each other following sub-clustering (Supplementary Fig. 1c, d). The expression of individual genes within clustered cells was then denoised and imputed using Markov affinity-based graph imputation of cells (MAGIC)[32].

In scRNA-seq UMAP projections, the distances between cells reflect the degree of transcriptional similarity between those cells, and UMAP visualization indicated relatively similar localizations of secretory clusters (Paneth/goblet cells, tuft cells, and enteroendocrine cells) from crypts isolated during homeostasis and after transplantation (Fig. 1b). In contrast, clusters representing the enterocyte lineage appeared to segregate more distinctly between experimental conditions. In particular, the positioning of ISC clusters (starred in Fig. 1b) suggested substantial transcriptional divergence, with the greatest distance appearing between ISCs isolated during homeostasis and ISCs isolated after allo-BMT. These localizations in UMAP space suggested that GVHD and its alloreactive mucosal infiltration may have an outsized impact on the transcriptomes of ISCs. To assess this directly, we computed phenotypic distances between the cells in each experimental condition in high-dimensional transcriptomic space, thus quantifying the transcriptional variation of distinct epithelial populations (see Methods). Upon comparing populations in homeostasis, syn-BMT, and allo-BMT, the ISCs consistently demonstrated the greatest phenotypic divergence (Fig. 1d and Supplementary Fig. 1e). Therefore, despite exposure to the same pre-transplant conditioning prior to allo-BMT and syn-BMT, the stem cells appeared to undergo substantial transcriptomic change after the transfer of allogeneic T cells and the subsequent dysregulated immune response.

We next sought to evaluate the impact of a less fulminant immune response on the GI tract, isolating crypt single cells for scRNA-seq after minor histocompatibility antigen (miHA)-mismatched allo-BMT (LP-into-B6). As donor T cell activation and intestinal infiltration proceed more slowly in this model[17], the epithelium was evaluated at a slightly later timepoint to provide additional time for the allogeneic immune response to develop. Thus, in addition to normal B6 crypts in homeostasis, SI crypts were isolated ten days post-BMT, after either transplanting marrow and T cells to induce this less aggressive GVHD or transplanting negative control T-cell-depleted (TCD) marrow. Single cell transcriptomes were sequenced, annotated (Supplementary Fig. 2a–c), and subjected to phenotypic distance analysis (Fig. 1e and Supplementary Fig. 2d) as before. Once again, despite the use of a distinct model and timepoint for analysis, the ISCs demonstrated the greatest transcriptional change in response to allo-BMT and immune-mediated damage, whether comparing populations from GVHD crypts with normal crypts (Supplementary Fig. 2d) or comparing them with crypts isolated after TCD transplants lacking GVHD (Fig. 1e).

In order to investigate the driving forces behind the profound transcriptional change occurring in GVHD, ISC genes differentially expressed following syn-BMT and MHC-mismatched allo-BMT were identified using MAST (Model-based Analysis of Single-cell Transcriptomics)[33] (Supplementary Data 1) and evaluated by gene set enrichment analysis (GSEA). Comparing post-transplant ISC clusters, GSEA identified the Hallmark Interferon Gamma Response to be the pathway most highly associated with ISC transcriptomes in GVHD (Fig. 1f). Similarly, comparison of ISC clusters in the TCD and T-cell-replete miHA-mismatched datasets also identified the Hallmark Interferon Gamma Response to be the pathway most highly associated with ISC transcriptomes in this distinct model of GVHD (Supplementary Fig. 2e). STAT1 is a principal mediator and transcriptional target of IFNγ signal transduction[23], and it was the ISC cluster's most upregulated transcription factor in the setting of GVHD (Fig. 1g and Supplementary Data 1). Notably, in addition to their high expression of Stat1, ISCs demonstrated the greatest Stat1 upregulation when compared with

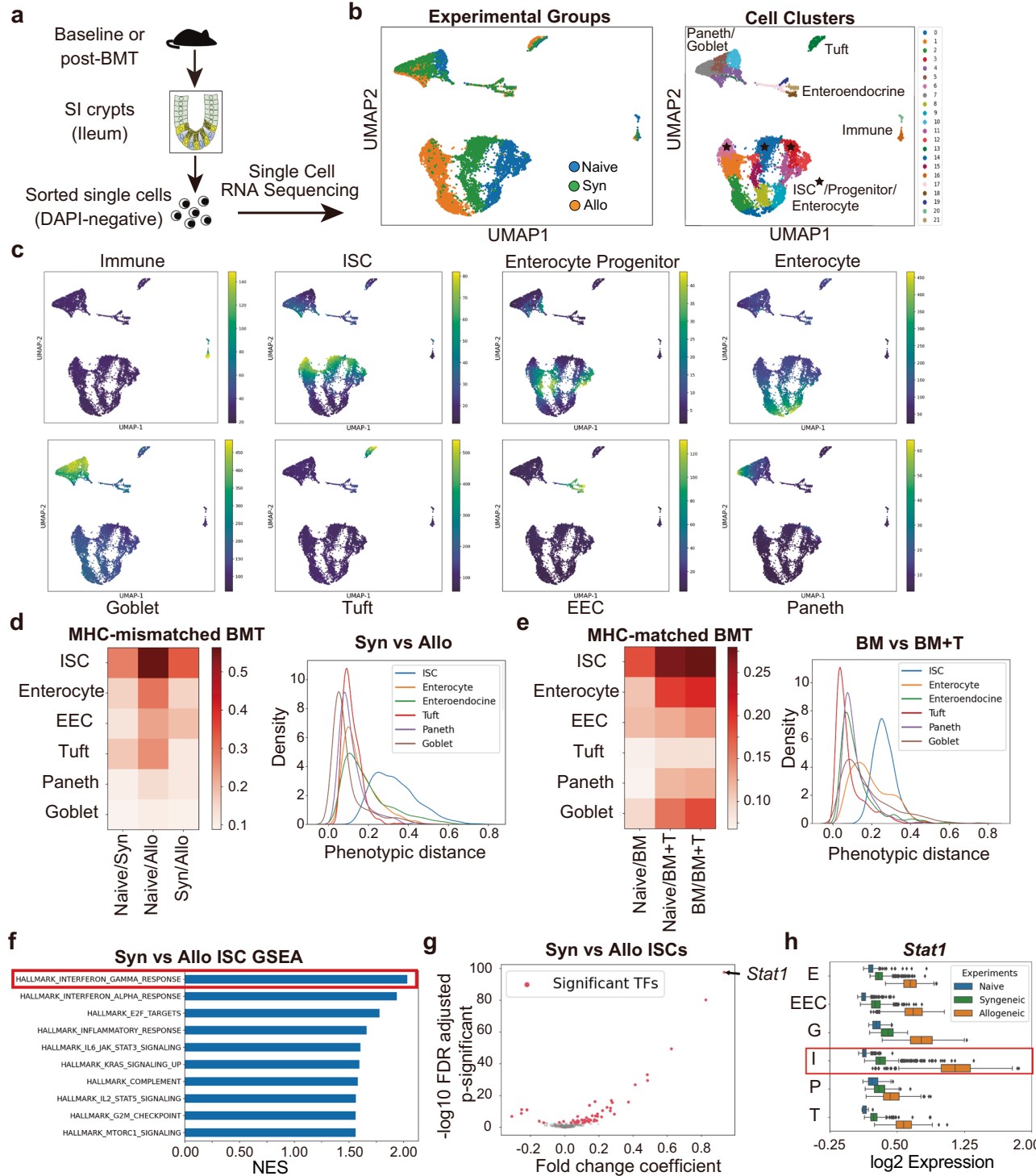

other intestinal epithelial populations after allo-BMT (Fig. 1h). Therefore, in two distinct BMT models analyzed at two distinct timepoints, ISCs demonstrated a substantial transcriptomic shift during GVHD that was highly associated with a response to the IFNγ/STAT1 axis.

## Epithelial STAT1 contributes to intestinal regeneration after bone marrow transplantation

We next sought to investigate the role of STAT1 in the intestines after damage by performing transplants into recipient mice lacking STAT1 in the intestinal epithelium. *Stat1*<sup>fl/fl</sup>×*Villin-Cre* (*Stat1*<sup>ΔIEC</sup>) mice, in which STAT1 is specifically deleted from intestinal epithelial cells, and Cre-

negative STAT1-intact "wild-type" (WT) *Stat1*<sup>fl/fl</sup> (*Stat1*<sup>WT</sup>) mice were thus utilized as transplant recipients for either syngeneic (B6-into-B6) or MHC-mismatched allogeneic (B10.BR-into-B6) BMT. Recipients were monitored post-transplant for their systemic responses, including weight loss and survival. Syn-BMT challenged both STAT1-intact and STAT1-deficient recipients with the same pre-transplant TBI conditioning and overall experimental procedures, while allo-BMT also introduced immune-mediated intestinal injury due to GVHD (Fig. 2a). Unexpectedly, in the presence of this immune-mediated damage, depletion of STAT1 from recipient intestinal epithelium failed to improve overall outcomes after allo-BMT, and STAT1-deficient

**Fig. 1 | Immune-mediated GI damage impacts the transcriptome of intestinal stem cells, upregulating STAT1 in GVHD. a–d, f–h** Single cell RNA sequencing (scRNA-seq) of freshly isolated ileal crypt cells from healthy B6 mice (Naive), B6-into-B6 syngeneic BMT recipients without GVHD (Syn), or B10.BR-into-B6 MHC-mismatched allogeneic BMT recipients with GVHD (Allo) 5 days after transplant. **a** Study design: Prior to scRNA-seq, BMT recipients were transplanted with both marrow and purified donor T cells following pre-transplant conditioning (TBI; 5.5 Gy x 2). **b** UMAP plots indicating the experimental conditions for sequenced cells (left) and distinct clusters with their cell type annotation (right). Star symbol indicates ISC clusters. **c** UMAP plots highlighting gene expression signatures corresponding to each cell type. **d** Phenotypic distances among epithelial populations. Heatmap shows average phenotypic distances between experimental conditions; histogram shows the distribution of phenotypic distances between non-GVHD and GVHD conditions (syn-BMT and allo-BMT) in units of log-normalized gene expression. **e** Phenotypic distances among epithelial populations from SI crypt

scRNA-seq after LP-into-B6 MHC-matched (minor-antigen-mismatched) allo-BMT in units of log-normalized gene expression. Heatmap shows each population's average phenotypic distance between homeostatic (Naive), non-GVHD T-cell-depleted transplant (BM), and GVHD-inducing BM + T cell transplant (BM + T); histogram shows the distribution of phenotypic distances between non-GVHD and GVHD transplant conditions (BM vs. BM + T). **f** GSEA comparing ISCs from syn-BMT and MHC-mismatched allo-BMT; NES: normalized enrichment score. **g** Volcano plot showing transcription factor genes differentially expressed (computed using MAST) in ISCs following syn-BMT and allo-BMT. Positive log fold change indicates increased expression after allo-BMT. **h** Expression of *Stat1* in crypt epithelial populations highlighting upregulation in ISCs after allo-BMT; gene expression imputed using MAGIC (n = 3685 Naive, 3371 Syngeneic, 4670 Allogeneic cells); E enterocytes, EEC enteroendocrine cells, G goblet cells, I ISCs, P Paneth cells, T tuft cells. The boxplots represent three quartiles, and the whiskers indicate 1.5 times the interquartile range.

recipients demonstrated more rapid mortality and greater weight loss (Fig. 2b, c).

Exploring the role of epithelial STAT1 at the tissue level, *Stat1*$^{\Delta IEC}$ and *Stat1*$^{WT}$ mice were examined histologically in the ileum after syngeneic and allogeneic BMT, and we observed a difference in the proliferative state of STAT1-deficient epithelium in GVHD. In the absence of alloreactive immune-mediated damage, both STAT1-deficient and STAT1-intact mice appeared histologically similar seven days after syn-BMT (Fig. 2d). This was consistent with their uniform survival following syngeneic transplantation (Fig. 2b). In the setting of GVHD after allo-BMT, STAT1-intact *Stat1*$^{WT}$ mice demonstrated an augmented regenerative response with increased Ki67$^+$ cells per crypt compared to the same *Stat1*$^{WT}$ mice after syn-BMT (Fig. 2d, e). However, intestinal STAT1-deficient mice demonstrated a reduced regenerative response with fewer Ki67$^+$ cells per crypt than the STAT1-intact recipients after allo-BMT (Fig. 2d, e). This reduction in Ki67 was still evident seven days later on day 14 post-BMT (Fig. 2f). Early after allo-BMT (day 7), there was a transient preservation of ISC frequencies in STAT1-deficient recipients (Supplementary Fig. 3), but by day 14 post-BMT, there were fewer crypt base Olfm4$^+$ ISCs in intestinal STAT1-deficient mice (Fig. 2g h). These findings thus indicated that intestinal STAT1 deficiency led to more rapid mortality after allo-BMT, reduced proliferation within the intestinal crypt epithelium, and ultimately attenuated ISC recovery.

As noted above, GSEA comparing ISC gene expression profiles after allo-BMT and syn-BMT identified the IFNγ response to be the pathway most associated with ISCs in GVHD. While this is consistent with the upregulation of *Stat1* observed in ISCs in GVHD (Fig. 1g, h), as *Stat1* itself is a target of IFNγ signaling, IFNγ is not the only cytokine that signals through STAT1. Additionally, GSEA comparing ISC clusters after allo-BMT and syn-BMT also identified the Hallmark Interferon Alpha Response as a potentially enriched pathway in GVHD (Fig. 1f), although there was limited evidence for upregulation of unique IFNα gene targets within this gene set (Supplementary Fig. 1f), suggesting that the identification of the IFNα pathway may have been driven by the IFNγ-associated genes included in the IFNα Hallmark gene set. To investigate experimentally if IFNα responses could be associated with the differences in epithelial proliferation observed in intestinal STAT1-deficient mice in GVHD, we performed syn-BMT and allo-BMT into WT and IFNα receptor (IFNαR)-deficient recipients. However, there were no changes in epithelial proliferation due to IFNαR deficiency after syn-BMT or allo-BMT (Fig. 2i), suggesting that Type 1 IFNs were not responsible for the epithelial regeneration associated with STAT1 post-BMT.

To examine the role of the IFNγ pathway, BMT was performed into recipients lacking the IFNγ receptor (IFNγR) (Fig. 2j). Similar to IFNαR deficiency in the setting of syn-BMT, there was no difference in crypt epithelial proliferation between IFNγR-floxed recipients with intact IFNγR signaling (*IFNγR*$^{WT}$) and IFNγR-deficient *Ifngr*$^{fl/fl}$x*Villin-Cre*

(*IFNγR*$^{\Delta IEC}$) recipients after syn-BMT. In contrast, while allo-BMT was again associated with increased crypt proliferation in WT recipients, *IFNγR*$^{\Delta IEC}$ recipients lacking IFNγR expression in the intestinal epithelium demonstrated a reduced proliferative epithelial response with fewer Ki67$^+$ cells per crypt than IFNγR-intact recipients. Reduced epithelial proliferation was also observed in *IFNγR*$^{\Delta IEC}$ recipients following miHA-mismatched (MHC matched) BMT in the LP-into-B6 GVHD model (Fig. 2k). Furthermore, loss of Olfm4$^+$ ISCs in GVHD was more severe in these *IFNγR*$^{\Delta IEC}$ recipients lacking IFNγR expression in the intestinal epithelium than it was in WT control BMT recipients (Fig. 2l). Therefore, STAT1 deficiency and IFNγR deficiency were both associated with reductions in the proliferative response of the intestinal epithelium in GVHD, which correlated with more severe reductions in crypt base ISCs.

## T cell-derived IFNγ stimulates STAT1-dependent epithelial proliferation

Given the IFNγ-associated *Stat1* response occurring in ISCs after allo-BMT (Fig. 1), the reduced epithelial proliferation present in STAT1-deficient and IFNγR-deficient BMT recipients (Fig. 2), and the known expression of IFNγ within the GI tract early post-transplant[23,24], we hypothesized that the IFNγ/STAT1 axis may contribute to regeneration of intestinal epithelial cells after BMT. Thus, we next examined the role of the IFNγ/STAT1 axis in epithelial regeneration more directly using an ex vivo intestinal organoid culture model. While co-culture with allogeneic T cells results in an IFNγ-dependent reduction in viable SI organoids, T cell co-culture also augmented the size of the surviving organoids, also in an IFNγ-dependent manner (Fig. 3a). Organoid size assessment can serve as a surrogate marker for epithelial proliferation[10], and human T cells also increased the sizes of genetically disparate human duodenal organoids, and anti-IFNγ treatment suppressed this increase in size as well (Fig. 3b and Supplementary Fig. 4a). We next examined the effects of IFNγ directly by supplementing cultures with recombinant IFNγ. Organoids that survived after IFNγ exposure exhibited significantly increased size as well as upregulation of the cell cycle regulator *Ccnd1* (Fig. 3c, d).

To investigate the proliferative effects of IFNγ on ISCs specifically, we used a niche-independent ISC culture model[34] composed nearly entirely of symmetrically dividing Lgr5$^+$ cells (Supplementary Fig. 4b, c). Despite the robust environment already present in the stem cell cultures due to enhanced Wnt and Notch signaling, addition of IFNγ further increased the growth of these ISC colonies (Fig. 3e and Supplementary Fig. 4d). Confirming the effects on epithelial proliferation indicated by *Ccnd1* expression and organoid and ISC colony size measurements, flow-cytometry-based cell cycle analysis of Lgr5-GFP$^+$ ISCs grown in standard EGF/Noggin/R-spondin-1-based intestinal organoid cultures from Lgr5-GFP mice demonstrated augmentation of ISC cell cycle progression to S/G2/M after IFNγ treatment (Fig. 3f and Supplementary Fig. 5).

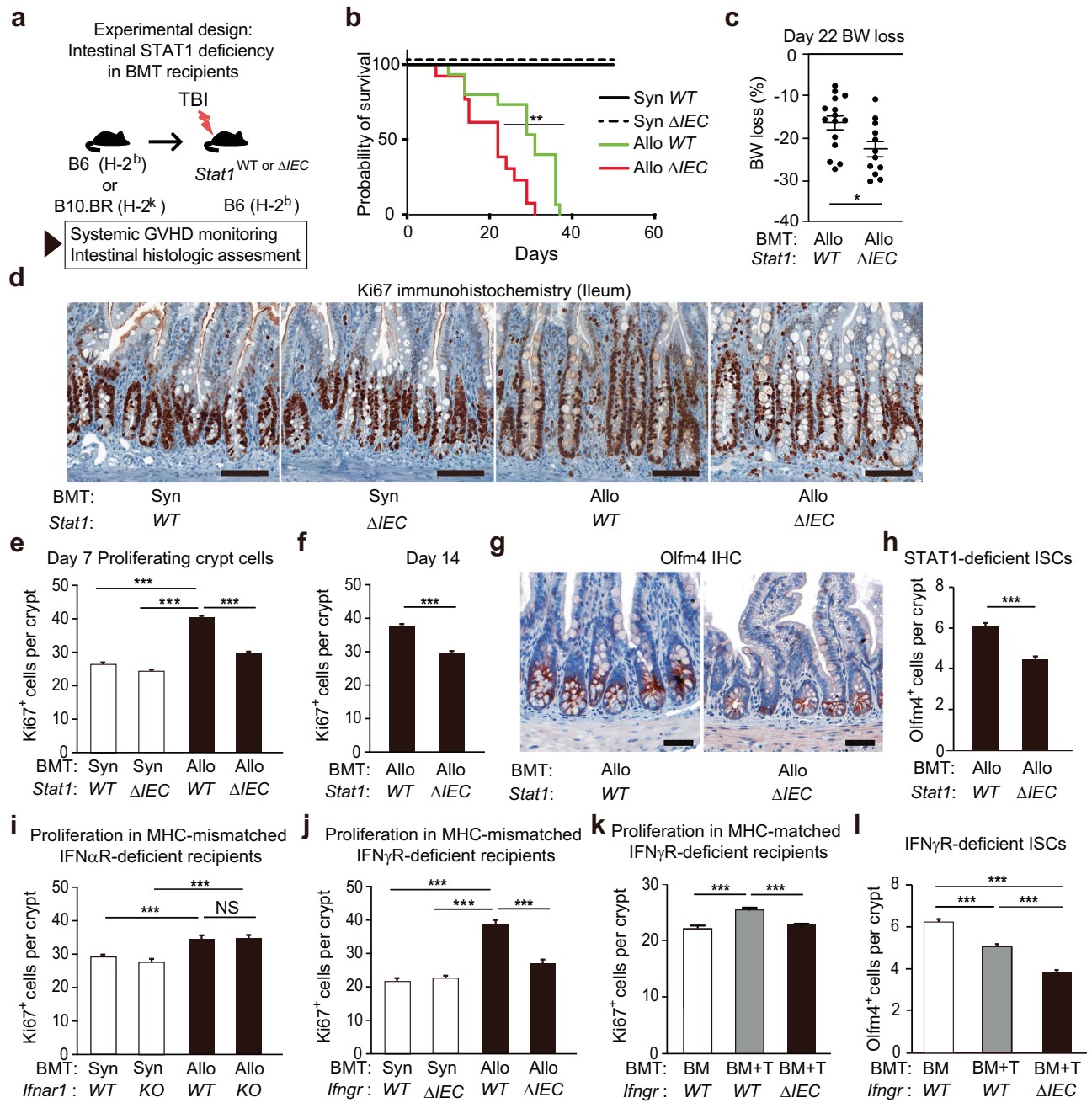

**Fig. 2 | Intestinal deficiency of STAT1 or IFNγR both reduce epithelial proliferation post-transplant. a–h** Allogeneic B10.BR-into-B6 (Allo) or syngeneic B6-into-B6 (Syn) BMT using *Stat1*fl/fl x*Villin-Cre* (*Stat1*ΔIEC) or Cre-negative *Stat1*fl/fl (*Stat1*WT) littermate controls. **a** Experimental design. **b** Survival curves; n = 3 Syn *Stat1*WT, 3 Syn *Stat1*ΔIEC, 15 Allo *Stat1*WT, 13 Allo *Stat1*ΔIEC mice; Log-rank-test (Allo *Stat1*WT vs Allo *Stat1*ΔIEC, *p* = 0.0034). **c** Body weight loss; n = 15 *Stat1*WT, 12 *Stat1*ΔIEC mice (*p* = 0.0179). **d-e** Representative staining (**d**) and quantification (**e**) of Ki67+ crypt cells, day 7 post-BMT (n = 168 Syn *Stat1*WT, 178 Syn *Stat1*ΔIEC, 426 Allo *Stat1*WT, 528 Allo *Stat1*ΔIEC crypts). **f** Ki67+ crypt cells 14 days post-BMT (n = 253 Allo *Stat1*WT, 408 Allo *Stat1*ΔIEC crypts). Representative staining (**g**) and quantification (**h**) of crypt Olfm4+ cells 14 days post-BMT (n = 388 Allo *Stat1*WT, 703 Allo *Stat1*ΔIEC crypts; scale bars = 50μm). **i** Analysis of crypt Ki67 immunohistochemistry after B10.BR-into-B6 or B6-into-B6 BMT using IFNαR-intact (WT) or IFNαR−/− knockout (KO) recipients (n = 98 Syn *IFNαR*WT, 86 Syn *IFNαR*KO, 83 Allo *IFNαR*WT, 61 Allo *IFNαR*KO crypts; Allo

*Ifnar1*WT vs Allo *Ifnar1*ΔIEC, *p* = 0.9835). **j–l** Analyses after BMT into *Ifngr*fl/fl x*Villin-Cre* (*IFNγR*ΔEC) recipients or Cre-negative *IFNγR*fl/fl (*IFNγR*WT) littermate controls. **j** Analysis of crypt Ki67 immunohistochemistry after B10.BR-into-B6 or B6-into-B6 BMT (n = 175 Syn *IFNγR*WT, 106 Syn *IFNγR*ΔIEC, 195 Allo *IFNγR*WT, 318 Allo *IFNγR*ΔIEC crypts). Frequencies of crypt Ki67+ cells (**k**; n = 248 BM *IFNγR*WT, 351 BM + T *IFNγR*WT, 361 BM + T *IFNγR*ΔIEC crypts) and frequencies of crypt Olfm4+ cells (**l**; n = 585 BM *IFNγR*WT, 978 BM + T *IFNγR*WT, 1053 BM + T *IFNγR*ΔIEC crypts) 10 days after LP-into-B6 BMT with either donor marrow alone (BM) or marrow and T cells (BM + T). Panels (**b**, **c**, **e**, **f**, **h**, **k**, **l**) are combined from two independent experiments; (**d**, **g**, **j**) are representative of two independent experiments. Graphs indicate mean and s.e.m.; comparisons performed with two-tailed t tests (two groups) or one-way ANOVA multiple comparison testing (multiple groups); * *p* < 0.05, ** *p* < 0.01, *** *p* < 0.001. The exact p values are *p* < 0.001 unless specified otherwise.

While IFNγ increased proliferation in intestinal organoids, it remained possible that this was a secondary response to IFNγ-mediated injury, rather than a direct induction of epithelial proliferation. To examine if this proliferative response was injury-dependent, we titrated down the concentration of IFNγ to identify non-toxic levels that did not reduce organoid frequencies. Decreasing the IFNγ concentration to 0.01 ng/ml left organoid numbers largely intact, and organoids survived throughout the culture period

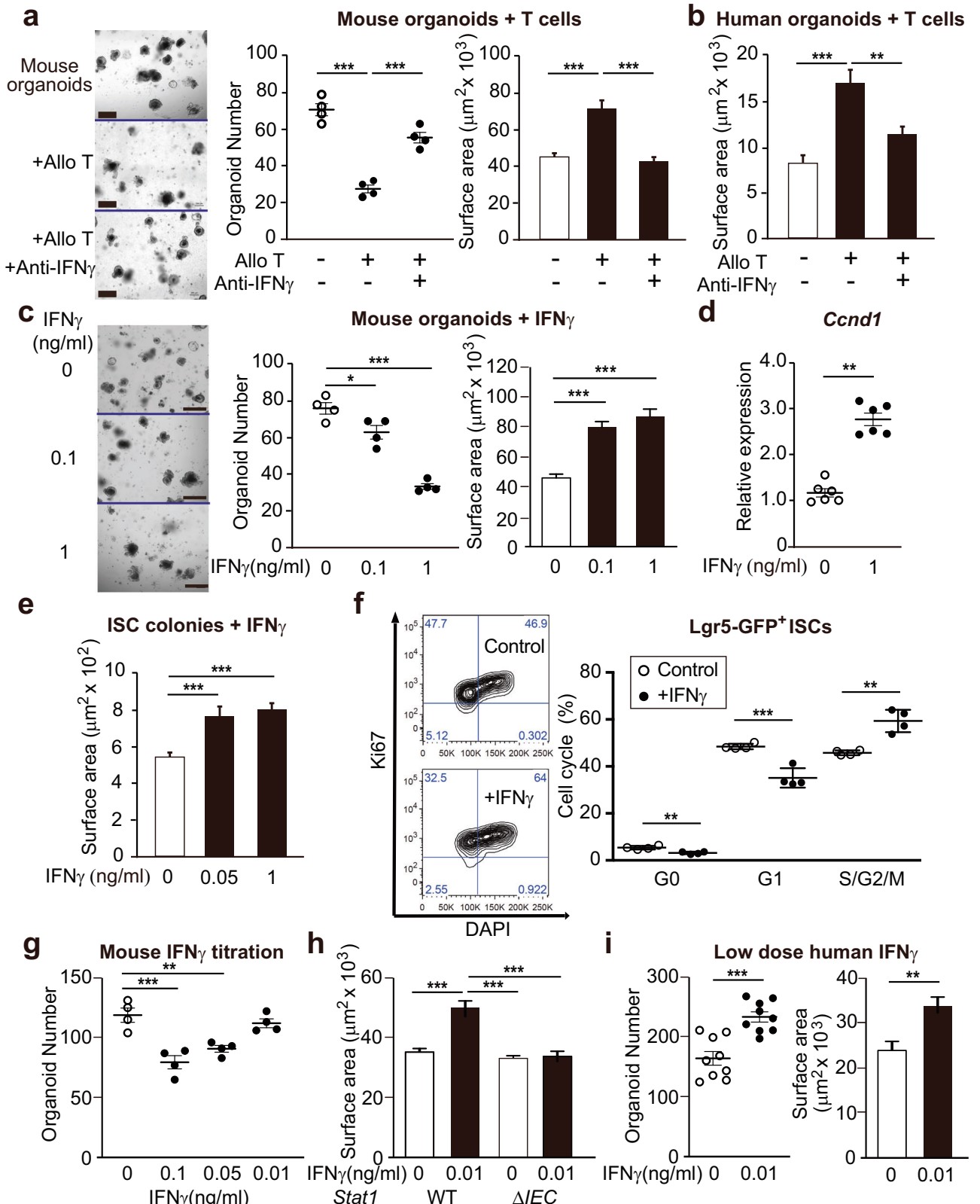

(5 days) at a similar proportion to those grown in standard conditions without IFNγ (Fig. 3g). Despite the reduction in toxicity, exposure to low-dose IFNγ continued to augment organoid size, and it did so in a STAT1-dependent manner (Fig. 3h). Human duodenal organoids also continued to demonstrate an increase in size in response to the lower concentration of IFNγ, and this growth augmentation

also extended to organoid frequency, with increased numbers of viable human organoids growing in the presence of low-dose IFNγ (Fig. 3i and Supplementary Fig. 4e). In total, these results indicated that T cells can promote epithelial regeneration via the IFNγ/STAT1 axis, which directly enhanced cycling of ISCs in ex vivo cultures.

**Fig. 3 | T-cell-derived IFNγ induces STAT1-dependent epithelial regeneration.**
**a** Representative images and quantification of B6 SI organoids co-cultured with allogeneic BALB/c T cells ± anti-IFNγ neutralizing antibodies; culture day 7; frequency: n = 4 wells/group; size: n = 152 (control), 64 (+ T cells), 114 (+ anti-IFNγ) organoids/group; scale bars = 500 μm. **b** Size of human SI organoids cultured 1:1 with human allogeneic T cells (500 single cells with 500 T cells) ± anti-IFNγ neutralizing antibodies; culture day 7; n = 75 (control), 107 (+ T cells), 99 (+ anti-IFNγ) organoids/group (T cells vs anti-IFNγ, p = 0.0011). **c** Representative images and quantification of SI organoids ± rmIFNγ for 7 days; n = 139 (0 ng/ml), 76 (0.1 ng/ml), 44 (1 ng/ml) organoids/group; scale bars = 500 μm (0 ng/ml vs 0.1 ng/ml, p = 0.0292). **d** *Ccnd1* qPCR of mouse SI organoids ± rmIFNγ for 24 h; n = 6 wells/group; two-tailed Mann–Whitney analysis (p = 0.0022). **e** Size of SI ISC colonies ± rmIFNγ for 3 days; n = 69 (0 ng/ml), 31 (0.1 ng/ml), 73 (1 ng/ml)

colonies/group. **f** Cell-cycle analysis of Lgr5-GFP$^{high}$ cells in SI organoids ± rmIFNγ (0.1 ng/ml) for 24 h; n = 4 wells/group (Control vs rmIFNγ, p = 0.0030 for G0; p = 0.0014 for S/G2/M). **g** Number of SI organoids +/- decreasing concentrations of IFNγ (culture day 5, n = 4 wells/group; 0 ng/ml vs 0.05 ng/ml, p = 0.0053). **h** Size of *Stat1*$^{WT}$ or *Stat1*$^{ΔIEC}$ B6 SI organoids ± "low dose" IFNγ [culture day 5; n = 132 (*Stat1*$^{WT}$), 110 (WT + IFNγ), 112 (*Stat1*$^{ΔIEC}$), 95 (ΔIEC + IFNγ) organoids/group]. **i** Quantification of human duodenal organoids +/- "low dose" IFNγ; culture day 7; frequency: n = 9 wells/group; size: n = 100 (0 ng/ml), 144 (0.01 ng/ml) organoids/group, p = 0.0011. Graphs indicate mean and s.e.m.; comparisons performed with two-tailed t tests (two groups) or one-way ANOVA multiple comparison testing (multiple groups), unless stated otherwise; ** p < 0.01, *** p < 0.001. The exact p values are p < 0.001 unless specified otherwise. Data are representative of five (**a**), two (**b**–**e**, **g**–**i**), or three (**f**) independent experiments.

## Stem cell Myc expression increases in response to immune-mediated GI damage

We next investigated how IFNγ signaling through tissue STAT1 could regulate epithelial regeneration. To gain a transcriptome-wide perspective, ISC clusters from scRNA-seq were interrogated to identify genes correlating with expression of *Stat1* post-transplant (see Methods, Supplementary Data 2). GSEA pathway analysis of genes correlating more closely with ISC *Stat1* expression after allo-BMT than after syn-BMT highlighted activation of the Myc (c-Myc) pathway in GVHD (Fig. 4a). While there is limited understanding of its physiologic function in normal/healthy adult intestines, Myc has been proposed to be a key downstream target of Wnt signaling in the ISC compartment, regulating epithelial proliferation and differentiation[35–37], and a contributor to ISC recovery from radiation injury[38–40]. However, the role of epithelial Myc in response to immune-mediated GI damage is unknown. Evaluation of Myc expression by scRNA-seq within distinct epithelial clusters indicated greater Myc expression in ISCs than in other crypt populations at baseline and after transplant (Fig. 4b). Following TBI pre-transplant conditioning and syn-BMT, ISC Myc expression appeared similar to the baseline expression present in normal ISCs. However, following BMT with allogeneic donor T cells inducing GVHD, Myc expression in ISCs was substantially elevated (Fig. 4b).

Further exploring the relationship between Myc and immune-mediated GI damage post-transplant, scRNA-seq indicated that elevated ISC Myc expression correlated highly with ISC expression of IFNγ-responsive genes and also with expression of *Stat1* (Fig. 4c). These correlations were evident in ISCs specifically in the GVHD setting after allo-BMT (Fig. 4c). Notably, these correlations were based on expression within individual cells, not just for the ISC population as a whole, suggesting an intimate connection between IFNγ and the epithelial Myc response, rather than an indirect Myc response in some cells after cytokine signaling in other cells. Additionally, this GVHD-associated increase in ISC Myc expression (Fig. 4b) coincided with the increased regeneration and Ki67 expression observed in surviving *Stat1*$^{WT}$ crypts early after allo-BMT (Fig. 2). Consistent with this observation, Myc expression in ISCs correlated highly with their expression of cell-cycle-related genes (Fig. 4d), potentially connecting IFNγ, STAT1, and the mucosal immune response to epithelial regeneration post-BMT.

To examine Myc expression at the protein level after BMT, we performed three-dimensional (3-D) immunofluorescent imaging of whole-mount ileal tissue following allo-BMT. MHC-mismatched recipients were transplanted with either donor marrow and T cells to induce GI GVHD or with TCD marrow on its own to serve as non-GVHD negative controls. The intestines were again isolated early post-BMT to examine the events driving the ensuing regenerative response, and the specimens were stained with anti-Olfm4 antibodies to identify ISCs, DAPI to identify nuclei, and anti-c-Myc. Imaging of the ISC compartment in the lower crypt region of the ileum indicated the presence of c-Myc expression, which co-localized

with Olfm4 staining (Fig. 4e). DAPI$^+$ cells in this region that were negative for Olfm4 likely represented Paneth cells, and they did not stain positive for Myc, consistent with gene expression identified by scRNA-seq (Fig. 4b). Also consistent with scRNA-seq, ileal crypts from recipients of allogeneic donor T cells demonstrated greater Myc expression than crypts from non-GVHD "BM Only" recipients (Fig. 4e). This was also apparent in imaging of the very bottom of the crypt (Fig. 4f). Olfm4 expression was abundant at the crypt base in non-GVHD recipients transplanted without donor T cells, and Olfm4 expression was reduced in GVHD mice transplanted with donor T cells. However, in the absence of GVHD these more numerous crypt base ISCs demonstrated less Myc protein expression than the persisting ISCs in GVHD mice transplanted with allogeneic donor T cells (Fig. 4f). Therefore, Myc protein expression in vivo was consistent with transcriptomic findings identified by scRNA-seq.

## IFNγ directly induces Myc-dependent intestinal regeneration

Based on in vivo and ex vivo findings, the IFNγ/STAT1 axis was associated with epithelial proliferation and ISC Myc expression following immune-mediated GI damage. To explore this relationship in more detail and determine direct effects of IFNγ signaling on individual epithelial populations, a scRNA-seq timecourse evaluation was performed on intestinal organoids exposed to IFNγ for 6, 24, and 48 h (Fig. 5a). All culture conditions were analyzed together, and cluster identities were annotated based on known lineage-defining expression patterns (Supplementary Fig. 6a, b). Consistent with the in vivo setting (Fig. 4b), baseline *Myc* expression was higher in ISCs than in differentiated epithelial cells (Fig. 5b). Following exposure to IFNγ, ISC *Myc* expression rapidly increased further, peaking within 6 h. Only a modest change in *Myc* expression was observed in enterocyte lineage cells and there was no impact on *Myc* expression within secretory lineages (Fig. 5b).

Direct and rapid induction of *Myc* expression was validated in IFNγ-treated SI organoids using qPCR (Fig. 5c). IFNγ exposure quickly induced expression of *Irf1*, a representative STAT1 target gene in the IFNγ signaling pathway. Myc upregulation coincided with this early phase of the IFNγ response, where it peaked as early as three hours after treatment. In contrast, although Myc is known to be a target of the Wnt pathway, expression of the representative Wnt target gene *Axin2* steadily decreased (Fig. 5c), indicating that activation of the Wnt pathway was unlikely be driving *Myc* upregulation in this setting. Additionally, while *Paf* has been reported to contribute to ISC *Myc* expression after radiation injury[39], *Paf* expression did not change in organoids or ISC colonies after IFNγ treatment (Supplementary Fig. 6c), suggesting that *Paf* may not be a key driver of *Myc* upregulation and regeneration here. Despite this and the apparent reduction in Wnt signaling, expression of the cell cycle regulator *Ccnd1*, a known downstream Myc target in ISCs[39], was upregulated in IFNγ-treated organoids (Fig. 5c), consistent with the increased organoid size and ISC cycling (Fig. 3). The kinetics of *Ccnd1* elevation were slightly delayed compared to *Myc* and *Irf1*, peaking six hours after exposure to IFNγ

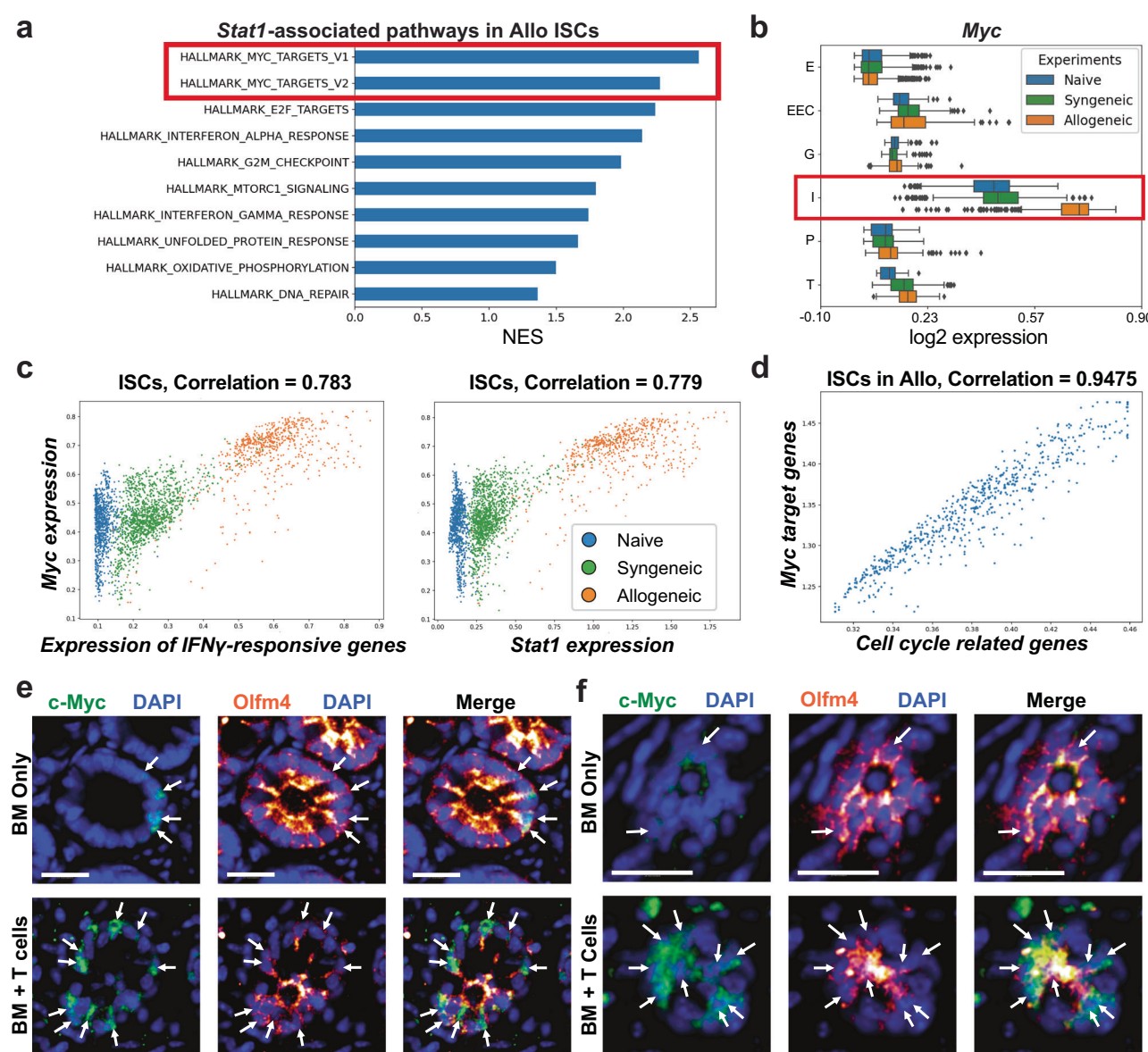

**Fig. 4 | Myc expression is elevated in crypt stem cells after allogeneic BMT. a–d**
Single cell RNA sequencing of small intestine crypt cells from healthy B6 mice
(Naive) or five days after B6-into-B6 syngeneic or B10.BR-into-B6 allogeneic BMT.
**a** GSEA of ISC cluster genes filtered for differential correlation with *Stat1* expression
in ISCs from syngeneic and allogeneic BMT recipients; NES: normalized enrichment
score. **b** *Myc* expression in crypt epithelial populations, highlighting *Myc* upregu-
lation in ISCs after allo-BMT; gene expression imputed using MAGIC; (n = 3685
Naive, 3371 Syngeneic, 4670 Allogeneic cells); E: enterocytes, EEC: enteroendocrine
cells, G: goblet cells, I: ISCs, P: Paneth cells, T: tuft cells. The boxplots represent
three quartiles, and the whiskers indicate 1.5 times the interquartile range.
**c** Correlation of *Myc* expression with expression of IFNγ-responsive genes or with

expression of *Stat1* in ISCs during homeostasis and after syngeneic or allogeneic
BMT. **d** Correlation between average MAGIC-imputed expression of Myc target
genes (Hallmark MYC_v1 and v2 pathways) and average imputed cell-cycle-related
genes (KEGG cell cycle pathway) in ISCs after allogeneic BMT. The plot shows
average gene expression (2nd–98th percentiles). **e-f** Imaging of full-thickness ileum
by 3-D whole mount microscopy four days after B10.BR-into-B6 allogeneic BMT
using either TCD BM alone, which does not result in GVHD, or BM and T cells,
resulting in GVHD. Shown are 2-D optical slices of the ISC compartment in the lower
crypt region (**e**) or the very base of the crypt (**f**) from 3-D fluorescent imaging
performed after staining with anti-c-Myc (green), anti-Olfm4 (orange glow), and
DAPI (blue). Arrows indicate c-Myc⁺Olfm4⁺ ISCs; scale bar: 25 µm.

(Fig. 5c). Strikingly, similar expression patterns were observed in
mouse ISC colonies evaluated following a timecourse of IFNγ exposure
(Fig. 5d) and in human duodenal organoids treated with IFNγ for six
hours (Fig. 5e).

In order to evaluate the functional importance of Myc in IFNγ-
driven epithelial regeneration, intestinal organoids were cultured in
the presence of the Myc inhibitor 10058-F4. Culture with 10058-F4
reduced the numbers and size of intestinal organoids in a
concentration-dependent manner, and higher concentrations were
highly toxic, eliminating viable organoid growth (Fig. 5f, g). While

viability was maintained at low concentrations of 10058-F4, presence
of the Myc inhibitor abrogated the impact of IFNγ on epithelial
regeneration, inhibiting IFNγ-driven organoid growth (Fig. 5h, i) and
*Ccnd1* induction (Fig. 5j).

To gain perspective on the dynamic contributions of Wnt signal-
ing and IFNγ/STAT1 signaling to Myc function, to quantify the rela-
tionship between Myc and these pathways, and to investigate the
in vivo relevance of reduced *Axin2* expression in IFNγ-treated orga-
noids, we performed a DREMI (conditional-Density Resampled Esti-
mate of Mutual Information) analysis[32,41] on scRNA-seq ISC clusters

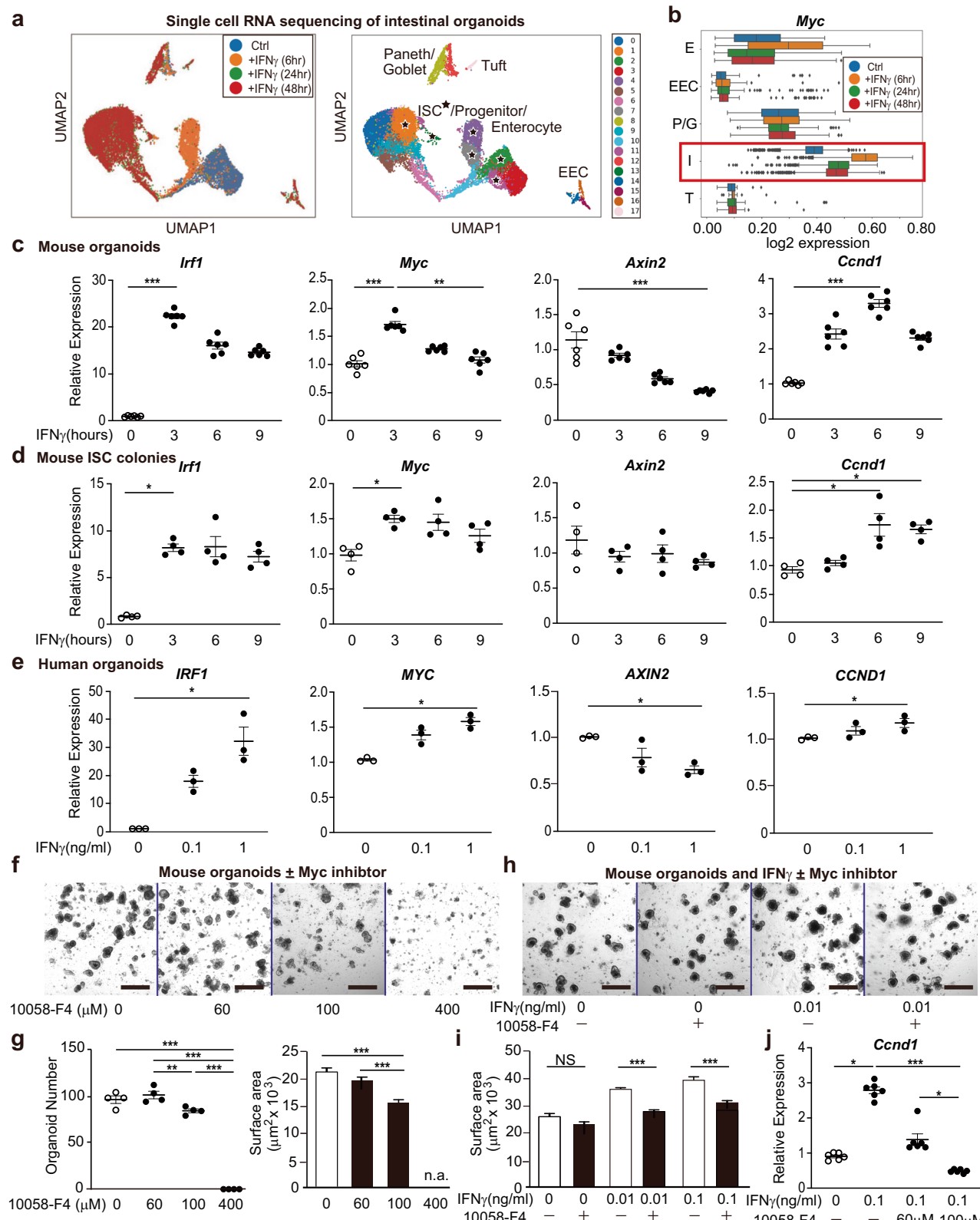

from BMT recipient mice (see Methods). This DREMI analysis computed the strength of gene-gene interactions for Myc target genes with Wnt pathway targets (Supplementary Data 3 and 4) and for Myc target genes with IFNγ targets (Supplementary Data 5 and 6) within ISCs post-transplant. These ISC gene-gene interactions were evaluated both after syn-BMT and after allo-BMT, and they indicated a stronger correlation between Myc and Wnt target genes after syn-BMT, while suggesting a stronger correlation between Myc and IFNγ target genes in the setting of immune-mediated GI damage after allo-BMT (Supplementary Fig. 6d). Therefore, while Wnt signaling may remain important for epithelial regeneration after damage, as observed in the setting of DSS-induced intestinal pathology[42], it appears to be less associated with the Myc pathway in the setting of immune-mediated GI damage due to GVHD.

**Fig. 5 | Myc is required for IFNγ-induced epithelial regeneration. a, b** Single cell RNA sequencing of B6 small intestine organoids after exposure to IFNγ; E: enterocytes, EEC: enteroendocrine cells, P/G: Paneth/goblet cells, I: ISCs, T: tuft cells. **a** UMAP plots indicating the experimental culture conditions and showing the cell type annotations for each cluster. Star symbol indicates ISC clusters. **b** *Myc* expression: n = 4599 (Ctrl), 3166 (6 h), 4533 (24 h), 6353 (48 h) cells. The boxplots represent three quartiles, and the whiskers indicate 1.5 times the interquartile range. qPCR analysis of mouse organoids (**c**, n = 6 wells/group) or ISC colonies (**d**, n = 4 wells/group) ± IFNγ (1 ng/ml) for gene expression of *Irf1* (d, 0 vs 3 h, *p* = 0.0360), *Myc* (c, 3 vs 9 h, *p* = 0.0045; d, 0 vs 3 h, *p* = 0.0227), *Axin2*, and *Ccnd1* (d, 0 vs 6 h, *p* = 0.0286; 0 vs 9 h, *p* = 0.0451); Kruskal-Wallis multiple comparison testing. **e** qPCR analysis of human organoids ± 6 h exposure to IFNγ (n = 3 donors/group); Friedman-tests; 0 vs 1 ng/ml, *p* = 0.0429 for *IRF1*; *p* = 0.0429 for *MYC*; *p* = 0.0429 for *AXIN2*; *p* = 0.0429 for *CCND1*. Representative images (**f**) and

quantification (**g**) of mouse organoids ± 10058-F4 (µM); culture day 4; frequency: n = 4 wells/group; size: n = 353 (0), 395 (60), 352 (100), 0 (400) organoids/group; one-way ANOVA multiple comparison testing; 60 vs 100 µM, *p* = 0.0089. Representative images (**h**) and quantification (**i**) of mouse organoids ± IFNγ (ng/ml) ± 10058-F4 (60µM); culture day 4; n = 251 (No IFNγ), 236 (10058-F4), 231 (IFNγ 0.01), 238 (IFNγ 0.01 + 10058-F4), 228 (IFNγ 0.1), 223 (IFNγ 0.1 + 10058-F4) organoids/ group; one-way ANOVA multiple comparison testing; No IFNγ vs 10058-F4, *p* = 0.1087. **j** qPCR for *Ccnd1* gene expression in mouse organoids ± IFNγ +/− 10058-F4 for 6 h; n = 6 wells/group; Kruskal-Wallis multiple comparison testing; No IFNγ vs 0.1 ng/ml, *p* = 0.0197; IFNγ 0.1 ng/ml + 10058-F4 60 vs 100 µM, *p* = 0.0197. Graphs indicate mean and s.e.m.; * *p* < 0.05, ** *p* < 0.01, *** *p* < 0.001. The exact *p* values are *p* < 0.001 unless specified otherwise. Scale bars = 500 µm. Panels (**c, d, f–g**) are representative of two independent experiments.

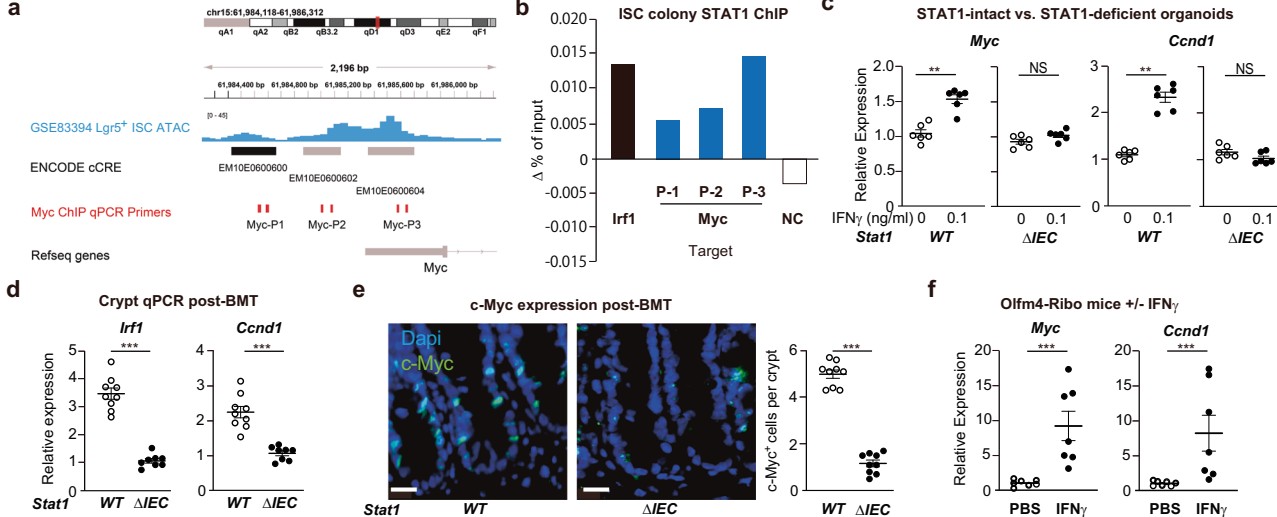

**Fig. 6 | STAT1 regulates epithelial *Myc* expression in immune-mediated GI damage. a** Diagram of the *Myc* gene locus (NCBI Refseq) with mapping of the genome location, ATAC-seq peaks (blue) from sorted Lgr5GFP ISCs (GSE83394), ENCODE (Accession EM10E0600598) candidate cis-regulatory elements (cCREs) indicating a proximal enhancer (black) and promoter regions (gray), and location of primers (red) used to detect STAT1 binding by qPCR following anti-STAT1 antibody and isotype control ChIP. **b** Δ% of input of anti-STAT1/isotype ChIP-qPCR of *Irf1*, *Myc* and negative control (NC) *Foxp3* loci performed on ISC colonies treated with IFNγ (3 h, 1 ng/ml). **c** *Myc* and *Ccnd1* qPCR analysis of mouse *Stat1*WT or *Stat1*ΔIEC SI organoids cultured ± IFNγ (n = 6 wells/group); *Stat1*WT vs IFNγ, *p* = 0.0022 for both *Myc* and *Ccnd1*; *Stat1*ΔIEC vs IFNγ, *p* = 0.0649 for Myc, *p* = 0.1255 for *Ccnd1*; combined from two independent experiments. **d, e** Allogeneic B10.BR-into-B6 BMT using *Stat1*ΔIEC recipients or *Stat1*WT littermate control recipients. **d** qPCR analysis of SI

crypts harvested day 7 post-BMT (n = 9 *Stat1*WT, 8 *Stat1*ΔIEC recipients; combined from two independent experiments). **e** Immunofluorescent staining and quantification of c-Myc (green) in ileum (day 5 post-BMT; n = 9 independent sections each from 3 *Stat1*WT and 3 *Stat1*ΔIEC recipients; scale bars = 20 µm; two-tailed t test). **f** Olfm4-Cre x RiboTag (Olfm4-Ribo) mice were treated intraperitoneally with 20µg IFNγ or PBS twice, 48 and 24 h prior to harvest, as well as tamoxifen 20 h prior to harvest. Hemagglutinin-labeled Olfm4+ ISC ribosomes were then isolated from small intestine, and qPCR was performed on the associated RNA transcripts; n = 7 mice/group; combined from three independent experiments. Graphs indicate mean and s.e.m.; comparisons performed with two-tailed Mann–Whitney unless otherwise specified; ** *p* < 0.01, *** *p* < 0.001. The exact p values are *p* < 0.001 unless specified otherwise.

## STAT1 regulates immune-mediated induction of intestinal Myc expression

Given the importance of STAT1 for epithelial proliferation in vivo after allo-BMT (Fig. 2) and the effects of IFNγ on regeneration (Fig. 3) and Myc expression (Fig. 5) ex vivo, we interrogated the *Myc* promoter region to evaluate if IFNγ-induced STAT1 signaling could directly target the Myc locus in ISCs. With guidance from ATAC-Seq of Lgr5+ ISCs (GSE83394)[43], PCR primers were designed to target areas of open chromatin localizing to putative regulatory elements upstream of the *Myc* start site (Fig. 6a). Chromatin immunoprecipitation (ChIP) was then performed for STAT1 using ISC colonies treated with IFNγ, followed by qPCR for genomic regions corresponding to predicted regulatory elements. To evaluate STAT1 binding, relative qPCR amplification was compared for chromatin isolated using anti-STAT1 vs. isotype antibodies. Positive control primers targeting the *Irf1* locus confirmed ChIP enrichment and STAT1 binding for this known IFNγ/STAT1 target, and negative control primers targeting the *Foxp3* locus

confirmed an absence of enrichment there (Fig. 6b). Moreover, qPCR amplification of the *Myc* promoter region after anti-STAT1 ChIP indicated enrichment for predicted regulatory elements at three upstream sites (Fig. 6b), suggesting direct binding of STAT1 to the *Myc* locus in ISCs exposed to IFNγ.

Next, SI organoids were evaluated to determine the functional significance of STAT1 for Myc expression. While IFNγ-treated WT organoids demonstrated upregulation of both *Myc* and the MYC target *Ccnd1*, STAT1-deficient organoids failed to augment expression of either one in response to IFNγ (Fig. 6c). The relationship between Myc and STAT1 was then evaluated in vivo. Following allo-BMT into *Stat1*WT and *Stat1*ΔIEC recipients, crypts isolated post-transplant confirmed a reduction in canonical IFNγ signaling in *Stat1*ΔIEC recipients as demonstrated by decreased *Irf1* expression (Fig. 6d). Consistent with ex vivo modeling, crypt *Ccnd1* expression was reduced as well in *Stat1*ΔIEC BMT recipients compared to *Stat1*WT controls (Fig. 6d). Furthermore, immunofluorescent staining indicated MYC protein expression in

$Stat1^{WT}$ ileal crypts after allo-BMT, but crypt MYC staining appeared reduced in $Stat1^{\Delta IEC}$ mice (Fig. 6e).

Finally, we directly examined the relationship between IFNγ, Myc, and ISC proliferation in vivo using Olfm4-Cre x RiboTag (Olfm4-Ribo) mice treated with IFNγ. Tamoxifen treatment of Olfm4-Ribo mice induces hemagglutinin labeling of ribosomes within Olfm4$^+$ cells. These labeled ribosomes and their associated mRNA transcripts can then be isolated by anti-hemagglutinin immunoprecipitation to measure in vivo ISC gene expression[44]. After administering two doses of IFNγ in vivo 24 h apart, assessment of gene expression 24 h after the last dose indicated ISC upregulation of both *Myc* and *Ccnd1* (Fig. 6f), consistent with findings from mice with GVHD (Fig. 4) and findings from IFNγ treatment ex vivo (Fig. 5). Therefore, in vivo, IFNγ administration on its own appeared sufficient to upregulate Myc expression and induce transcriptional changes associated with ISC proliferation. Taken altogether, the results from this study support a model of IFNγ/ STAT1-driven Myc expression and epithelial regeneration induced by T cells during the intestinal injury occurring in GVHD.

## Discussion

In order to investigate the intricate relationship between immune-mediated tissue damage, stem cell loss, and epithelial renewal, and how this disrupts the normal process of homeostatic epithelial maintenance in the GI tract, we performed scRNA-seq transcriptomic analysis of crypt epithelium after BMT. Upon analyzing all crypt constituents concurrently in this unsupervised fashion, the greatest transcriptomic change occurring in response to the dysregulated mucosal immune response was evident within the tissue stem cells. Crypt injury and regeneration are hallmarks of GI GVHD pathology, and the findings presented here reveal the role of immune-mediated activation of ISC STAT1 in this process. While intestinal regeneration is at times thought to occur as a reflexive tissue-intrinsic response to damage, and inflammatory cytokines are well known contributors to GI damage in GVHD, IFNγ was found to be a contributor to epithelial regeneration in this context. Rather than merely an inducer of pathology that indirectly resulted in a subsequent regenerative response, IFNγ was a direct inducer of stem cell Myc expression and proliferation. The duality of IFNγ, as a direct inducer of both toxicity and intestinal regeneration, appears to be driven by the amount of IFNγ and STAT1 signaling that a stem cell experiences, as high concentrations of IFNγ directly induce Bax/Bak-dependent ISC apoptosis[23], while low concentrations were able to promote Myc expression and ISC proliferation without inducing demonstrable toxicity. These unexpected findings were obtained as a result of the single cell methodology utilized for this project, allowing the unique response of the stem cell compartment and the distinct responses of individual stem cells to be distinguished from their neighbors and the complex multicellular environment surrounding them.

Overall, this study supports a model of ISC regulation whereby Myc expression may largely be associated with the Wnt pathway during homeostasis, but in the setting of immune-mediated GI damage occurring in GVHD, T cells and IFNγ-driven STAT1 signaling can play a prominent role in regulating this stem cell program (Fig. 7). While the immune system is known to impact the ISC compartment[10,11,17,30], this would indicate that the lymphocyte response is not restricted to having modulatory effects on tissue stem cells, such as inducing their toxicity or shifting their behavior, but can actually regulate a fundamental component of stemness itself[45]. Initially identified in the setting of tissue damage occurring in GVHD, STAT1-mediated regulation of ISC Myc was validated in non-transplant ex vivo organoid culture models, and this immune-mediated regulation of ISCs may have relevance for tissue responses to lymphocytes and inflammation in other settings as well. The role of IFNγ as a direct driver of epithelial proliferation and regulator of stem cell *Myc* expression illustrates the pleiotropic nature of this complex cytokine, countering its classical

**Fig. 7 | Proposed schematic model of ISC Myc regulation.** While regulation of ISC Myc expression may be primarily Wnt-driven at baseline, we propose that there is additional STAT1-dependent regulation of Myc expression driven by IFNγ in the setting of GVHD.

characterization as a cytostatic anti-tumor effector and providing a potential explanation for conflicting experimental and translational findings, including results in the setting of malignancy[46].

In the allo-BMT setting, the majority of IFNγ$^+$ cells in recipient intestinal mucosa have been identified as donor T cells, with a CD4$^+$ compartment demonstrating a T-bet$^+$ Th1 phenotype[23]. Although Type 2 and Type 3 immune responses include now well-appreciated roles in tissue maintenance and regeneration, for example through production of tissue-targeted cytokines such as Amphiregulin and IL-22, Th1 immunity has frequently been described without such potential[10–16]. Despite the typical characterization of Type 1 responses as physiologic inducers of pathogen clearance and pathophysiologic inducers of tissue damage, the demonstration that STAT1 can promote epithelial proliferation in response to IFNγ indicates that Th1 immunity can promote tissue regeneration as well. All three major branches of lymphocyte polarization may thus have tissue reparatory potential, perhaps physiologically being activated as necessary within the larger context of a given immune response that must utilize the most appropriate type of immunity to respond to a particular pathogen or other insult.

While these findings broaden the understanding of IFNγ biology to include promotion of epithelial proliferation in a STAT1-dependent manner, IFNγ is not the only cytokine known to activate STAT1. For example, type I IFN can also signal through STAT1, often as a heterodimer with STAT2, whereas IFNγ utilizes a distinct receptor to signal through STAT1 homodimers and also drive *Stat1* expression as shown here. Regardless, it is possible that other cytokines may be able to activate this STAT1-dependent regenerative axis as well, although type I IFN did not demonstrate that capacity here. Rather, the findings presented here demonstrate that IFNγ and IFNγR are necessary to promote this epithelial proliferation in vivo in the setting of GVHD and that they are sufficient to directly induce ISC Myc expression and proliferation in a STAT1-dependent fashion. How IFNγ manages to accomplish this while also being capable of inducing STAT1-dependent ISC apoptosis[23] is another example of its complex pleiotropic behavior. Our findings indicate that IFNγ concentration and signaling intensity may contribute to the distinct responses observed. In the experimental BMT setting, we speculate that IFNγ-dependent ISC killing may occur mainly early post-transplant when immune-mediated inflammation and cytokine storms flare in association with high IFNγ expression, while IFNγ-dependent ISC proliferation may play an important role in the regeneration necessary for epithelial maintenance during such injury and afterward as the inflammation is subsiding. Given the potential contributions to both tissue pathology and tissue recovery, investigating the impacts of IFNγ signaling and IFNγ-driven epithelial proliferation on intestinal barrier function is an important consideration for future work.

Finally, in addition to pathophysiology, these findings have therapeutic implications as well. If even pathologic Th1 immunity can contribute to tissue regeneration, clinical efforts to inhibit immune-mediated disease with JAK inhibitors, which are increasing in use[47], may possibly have unintended detrimental consequences in some

settings, potentially increasing tissue damage by blunting the regenerative arm of the immune response. This is perhaps most evident from the finding that intestinal STAT1 deficiency increased mortality after allo-BMT. While the precise mechanism of this mortality is not entirely clear, STAT1 deficiency consistently reduced epithelial proliferation in the GI tract after allo-BMT, thereby indicating an attenuated regenerative response. Clinical inhibition of JAK/STAT signaling may also have impacts on tissues and systemic sequelae beyond the intended effects on lymphocytes and other hematopoietic cells. However, complementary strategies overcoming the suppression of epithelial regeneration will have the potential to provide an optimal therapeutic approach that addresses immune-mediated damage more comprehensively, suppressing pathologic immunity while concurrently stimulating tissue repair.

## Methods

### Mice

C57BL/6 (B6, H-2$^b$), LP (H-2$^b$), and B10.BR (H-2$^k$) mice were purchased from The Jackson Laboratory. B6 *lgr5-gfp-ires-CreERT2* (Lgr5-GFP) mice were provided by Hans Clevers. *Stat1*$^{fl/fl}$ mice were provided by Lothar G. Hennighausen and Christopher A. Hunter. *Stat1*$^{ΔIEC}$ mice were generated by mating *Stat1*$^{fl/fl}$ mice with Villin-Cre mice [B6.Cg-Tg(Vil1-cre)997Gum/J] from The Jackson Laboratory. *Ifngr*$^{ΔIEC}$ mice were generated by mating *Ifngr*$^{fl/fl}$ mice with Villin-Cre mice, both from The Jackson Laboratory. *Ifnar1*$^{-/-}$ mice were purchased from The Jackson Laboratory as well. Olfm4-Ribo mice were generated by crossing *Olfm4-IRES-eGFPCreERT2* (Olfm4-Cre) mice provided by Hans Clevers with B6N.129-Rpl22 (RiboTag) mice from the Jackson Laboratory.

All animal experiments were performed in accordance with the guidelines of the Memorial Sloan Kettering Cancer Center (MSKCC) Institutional Animal Care and Use Committee. Mice were housed in micro-isolator cages, up to five per cage, in MSKCC specific-pathogen-free facilities and received standard chow and autoclaved sterile drinking water. The facilities are maintained at a room temperature of 72 degrees (± one degree) Fahrenheit (22 degrees Celsius) and a light/dark cycle of 12 h each. The effects of sex differences on biologic outcomes were not specifically examined in this study. Sex as a variable was accounted for when feasible by including material from male and female transgenic mice in organoid and transplant experiments. Mice were used with median ages of 2–3 months. To adjust for differences in weight and intestinal flora of transplant recipients, among other factors, genetically identical age-matched mice were purchased from The Jackson Laboratory and then randomly distributed into different cages and across experimental groups by a non-biased technician who had no insight or information about the purpose or details of the experiment. The investigations assessing clinical outcome parameters were performed by non-biased technicians with no particular knowledge or information regarding the hypotheses of the experiments and no knowledge of the specifics of the individual groups.

### Bone marrow transplantation

Bone marrow transplant (BMT) procedures were performed as previously described[23]. Major histocompatibility antigen-mismatched allogeneic BMT (B10.BR into B6, H-2$^k$ into H-2$^b$), minor histocompatibility antigen-mismatched allogeneic BMT (LP into B6, H-2$^b$ into H-2$^b$), and syngeneic BMT (B6-into-B6) models were utilized for this study. Transplants utilizing only wild-type recipients were typically performed using female mice from The Jackson Laboratory as recipients for transplantation at an age of 8–10 weeks, whereas transplants utilizing transgenic recipients were performed using a mix of male and female mice. Recipient mice were conditioned for transplantation using 1100 cGy TBI split in 2 doses at 3–4 h intervals to reduce radiation-related toxicity. To obtain bone marrow cells from euthanized donor mice, the femurs and tibias were collected aseptically, and the bone marrow canals washed out with sterile media. Bone marrow

cells were depleted of T cells by incubation with anti-Thy 1.2 and low-TOX-M rabbit complement (Cedarlane Laboratories). Purity of the T-cell-depleted (TCD) bone marrow product was confirmed by T cell quantification using flow cytometry. T cell contamination was typically ~0.2% of all leukocytes after a single round of complement depletion. Donor T cells were isolated from splenocytes using Miltenyi anti-CD5 beads and MACS columns. Recipients typically received $5 \times 10^6$ TCD bone marrow cells with or without $1–4 \times 10^6$ T cells per mouse via tail vein injection.

### Crypt isolation and cell dissociation

Isolation of intestinal crypts and the dissociation of cells for flow cytometry analysis were performed as previously described[10]. In brief, after euthanasia with $CO_2$, either the entire small intestine (SI) or the terminal 10 cm of ileum was collected, opened longitudinally, and washed with PBS. The tissue was then incubated at 4 °C in EDTA (10 mM) for 20 min to dissociate the crypts. To isolate single cells from crypts, the pellet was further incubated in 1× TrypLE express (Gibco, Life Technologies) supplemented with 2 kU/ml DNase1 (Roche).

### Singe cell RNA sequencing of intestinal epithelial cells

Dissociated crypt cells or intestinal organoids were stained with DAPI and then sorted to purify DAPI-negative cells. The single cell RNA sequencing (scRNA-seq) of FACS-sorted cell suspensions was performed on a Chromium instrument (10X genomics) following the user guide manual CG00052 Rev E and Single Cell 3' Reagent Kit (v2). Each sample, containing approximately 5000 cells at a final dilution of 66–70 cells/μl was loaded onto the cartridge according to the manual. The viability of cells was between 83-99%, as confirmed with 0.2% Trypan Blue staining using a Countess II cell counter.

The individual transcriptomes of encapsulated cells were barcoded during the reverse transcription (RT) step. The resulting cDNA was purified with DynaBeads. This was followed by amplification with 14 cycles of PCR: 98 °C for 180 s, 14x (98 °C for 15 s, 67 °C for 20 s, 72 °C for 60 s), and 72 °C for 60 s. Next, 50 ng of PCR-amplified product was fragmented, A-tailed, purified with 1.2X SPRI beads, ligated to the sequencing adapters, and indexed by PCR: 98 °C for 45 s; 12x (98 °C for 20 s, 54 °C for 30 s, 72 °C for 20 s), and 72 °C for 60 s. The indexed DNA libraries were double-size purified (0.6–0.8X) with SPRI beads and sequenced on the Illumina NovaSeq S4 platform (R1 – 26 cycles, i7 – 8 cycles, R2 – 70 cycles or higher). Sequencing depth was 198 million reads per sample on average (31,900 reads per cell). The cell-gene count matrix was constructed using the Sequence Quality Control (SEQC) package[48]. Viable cells were identified on the basis of library size and complexity, whereas cells with >20% of transcripts derived from mitochondria were excluded from further analysis. After barcode correction and filtering, we recovered 11,427 unique transcripts per cell, with a median read per transcript of 1.83. For clarity, we will refer to each scRNA-seq experiment as follows:

- scRNA-seq of B6 ileal crypt cells isolated from normal mice or from transplant recipients 5 days after syngeneic (syn) or MHC-mismatched allogeneic (allo) BMT: Experiment 1
- scRNA-seq of SI crypt cells from normal B6 mice and from LP-into-B6 MHC-matched BMT recipient mice 10 days after receiving BM alone or BM and T cells: Experiment 2
- scRNA-seq of B6 SI intestinal organoids treated with IFNγ (0.05 ng/ml for 6, 24, or 48 h; harvested for analysis on culture day 5): Experiment 3

### Processing of single cell RNA sequencing data

As noted above, the fastq files generated from sequencing were processed with SEQC[48] using default parameters to obtain a count matrix for each experiment. To avoid biases in downstream analysis due to

lowly expressed genes, all genes expressed in less than 16 cells were removed from each experiment. The count data from each experiment was then library-size-normalized followed by log2 transformation with a pseudocount of 1. All processing was performed using the Python package Scanpy[49].

The normalized data were further evaluated and processed for removal of noisy cells. For this, we first computed the top 4000 highly variable genes (using the highly_variable_genes function of Scanpy, with flavor = 'seurat'), computed the top 40 principal components, and clustered the cells using PhenoGraph[27] (k = 30). This clearly revealed clusters that had substantially low total RNA expression and few genes uniquely expressed (cluster 6 for Experiment 1, cluster 8 for Experiment 2, and clusters 11 and 12 for Experiment 3). Since this way of clustering the cells and removing the noisy clusters takes the entire transcriptome into account, this approach for removing noisy cells from the data was preferred over using a threshold based on total RNA-counts. Thus, we removed these clusters from respective experiments and reanalyzed the cleaned data. We recomputed the highly variable genes (top 4000), followed by principal component analysis (PCA) for the top 40 components, and re-clustered the cells using PhenoGraph (k = 30). Visualization was performed using UMAP computed on principal components[26].

For sub-clustering of goblet and Paneth cells in Experiment 1, we followed a similar procedure. We first created a subset by selecting the cells that belonged to clusters 4, 5, 7, and 10 (Fig. 1b) and recomputed highly varying genes (top 1000) followed by PCA (top 40 components) and PhenoGraph on PCA (k = 15) to obtain the cell sub-clusters.

## Phenotypic distance analyses
To quantify the impact of tissue damage on each cell type, we computed phenotypic distances for epithelial cell types across each experimental condition. For this, we began with the PCA of the data computed on the selected highly variable genes (see above), and, for a given cell type, we then calculated the cosine distance between each pair of cells across two conditions. To ensure that we were measuring distances between phenotypically similar cells, we limited the number of cells that each individual cell would be compared to in the corresponding clusters from other experimental conditions. For a given cell in one condition, we only considered $k$ closest cells in the other condition, and vice versa. The results for each cell are presented as a distribution (using the kdeplot function of the Seaborn package in Python) or as an average heatmap (Fig. 1d, e and Supplementary Figs. 1e, 2d). We present the results at $k = 15$, but the results were also tested for various other $k$ values and found to be robust across a wide range of values for $k$.

## Denoising and imputing gene expression
While scRNA-seq has been profoundly useful in numerous biological contexts, a lingering problem is that of missing transcripts or "dropouts". Due to the dropout effect, it is difficult to establish covariance or correlation pattern between any pair of genes. Therefore, to circumvent this effect, we applied MAGIC[32] to denoise and impute normalized and log-transformed gene expression from each experiment. Briefly, MAGIC (Markov Affinity-based Graph Imputation of Cells) is a graph-based technique to denoise and impute single-cell gene expression. It relies on the assumption that genes are often co-regulated and form correlated network modules. This correlation pattern of genes allows cells with similar transcriptomes to be in proximity of each other in the phenotype space (defined by their gene expression pattern), even if they suffer from dropouts. Therefore, by identifying the local neighborhood of each cell, we can transfer information (that is, missing gene expression) within that local neighborhood to impute and denoise each cell's expression.

We follow the same computational procedure as described by van Dijk et al.[32]. Briefly, we begin by constructing a k-nearest neighbor graph ($k = 30$, Euclidean distance) of the cells on the principal components (top 40). This distance-based graph is then converted to an affinity matrix $W$ using an adaptive kernel defined as $W_{i,j} = \exp\left(-\frac{dist(x_i, x_j)}{\sigma_{i,j}}\right)$, where $x_i$ and $x_j$ represent two cells and $\sigma_{i,j}$ (representing the variance) is defined as the distance to the $ka^{th}$ neighbor from cell $x_i$ (ka = k/3 = 30/3 = 10). Thus, the kernel ensures that the cells closer to each other in the phenotype space have higher affinity and that the neighborhood of each cell adapts to the density of cells around it (as captured by $\sigma_{i,j}$; denser neighborhoods imply smaller $\sigma$). This affinity matrix is then symmetrized ($W = W + W\_transpose$), row-normalized to get a Markov or a transition matrix ($M$) of size $N$-by-$N$ ($N$ = number of cells), whose every row sums to 1. Each row represents the probability of a random walker stationed at the cell - (to be thought of as a state of the walker) represented by the row – hopping into another cell in one step. Cells that are closer to each other will have higher probability and cells that are far away from each other will have low (or zero) probability. This transition matrix can be raised to integer power ($t$) to explore the dynamics of the random walker in $t$ steps. On the other hand, left multiplying the original data matrix by the transition matrix (or transition matrix to a power $t$) allows gene expression information to be shared among similar cells (or in a local neighborhood of each cell). In other words, since each row sums to 1, transition matrix times the data matrix results in weighted average gene expression for each cell (where similar cells get higher weights). For our purposes, we raise the transition matrix to a power of $t = 3$. Thus, $imputed\_data = M^3 * log-transformed\_data$.

## Gene set enrichment analysis
Gene set enrichment analyses (GSEA) were performed with Hallmark pathways obtained from the MSigDb website (http://www.gsea-msigdb.org/gsea/msigdb/collections.jsp) using the fGSEA R-package (https://github.com/ctlab/fgsea). To perform GSEA on genes differentially expressed between ISCs after allo-BMT and ISCs after syn-BMT (Fig. 1f and Supplementary Data 1), we ranked the genes based on -log10 of the adjusted p values multiplied by the sign of log-fold change. The same strategy was used to compare BM against BM + T (Supplementary Fig. 2e).

To perform GSEA on the genes correlating more closely with ISC *Stat1* after allo-BMT than after syn-BMT (Fig. 4a), we ranked the genes based on their correlation with *Stat1* after subtracting the syn-BMT *Stat1* correlation score from the allo-BMT *Stat1*correlation score for each gene (see below).

### *Stat1* correlation analysis
To identify pathways associated with ISC *Stat1* expression in immune-mediated GI damage (GVHD), gene expression profiles were analyzed in ISC clusters from scRNA-seq performed after syn-BMT and after allo-BMT. ISC *Stat1* associations attributable to GVHD were determined by subtracting the *Stat1* correlation score for each ISC gene after syn-BMT from the *Stat1* correlation score for that gene after allo-BMT. Genes were then ranked based on the resulting (subtracted) correlation score (Supplementary Data 2) and subjected to GSEA pathway analysis. The ten highest ranking Hallmark pathways upregulated in allo-BMT are shown in Fig. 4a. The correlations were computed on the MAGIC-imputed data.

### DREMI analysis
In order to conceptualize the relationship between the Wnt, STAT1 and MYC pathways, we studied the association between candidate genes in each of these pathways using conditional-Density Resampled Estimate of Mutual Information (DREMI) analysis[32,41]. For this, we considered all the genes implicated in these pathways based on existing signatures associated with each of these genes, namely:'HALLMARK_WNT_BETA_CATENIN_SIGNALING','HALLMARK_INTERFERON_GAMMA_RESP

ONSE', and 'HALLMARK_MYC_TARGETS_V1' + 'HALLMARK_MYC_TARGETS_V2'.

These data were retrieved from the MSigDB website (http://www.gsea-msigdb.org/gsea/msigdb/). To conceptualize the relationship between these pathways, it was determined to be most effective to only consider genes that have substantial expression dynamics in the data such that they could contribute meaningfully to the average pathway expression. As such, we focused our analysis on genes that had a meaningful correlation with the average pathway gene expression. Therefore, we first computed the correlation of each gene in each of these signatures with the average gene signature expression. From this list, we filtered for genes that had a correlation score greater than 0.5 standard deviations above the mean. We then used these genes to perform pairwise DREMI analysis (Supplementary Data 3-6). The results were displayed as density functions (Supplementary Fig. 5d), and p values were computed using the RankSum test in the stats module within the Scipy package in Python. This computation was performed using the imputed data.

### Organoid and ISC colony culture

For mouse organoids, depending on the experiments, 100–200 crypts or 2000-3000 dissociated single organoid cells per well were suspended in growth-factor-reduced Matrigel (Corning) mixed with DMEM/F12 medium (Gibco). After Matrigel polymerization, complete ENR medium containing advanced DMEM/F12 (Sigma), 2 mM Glutamax (Invitrogen), 10 mM HEPES (Sigma), 100 U/ml penicillin, 100 μg/ml streptomycin (Sigma), B27 supplement (Invitrogen), N2 supplement (Invitrogen), 50 ng/ml mouse EGF (Peprotech), 100 ng/ml mouse Noggin (Peprotech) and 10% human R-spondin-1-conditioned medium from R-spondin-1-transfected HEK 293 T cells was added to small intestine crypt cultures. Media was replaced every 2–3 days. Along with medium changes, treatment wells received different concentrations of rmIFNγ (R&D systems) and/or 10058-F4 (Selleckchem). ISCs were isolated from Lgr5-GFP mice as above followed by several strainer steps and a 5-min incubation with TrypLE and 2 kU/ml DNase1 under minute-to-minute tapping to make a single-cell suspension.

ISC colonies were cultured from sort-purified single Lgr5-GFP+ cells. Approximately 3,000 ISCs were plated in 30 μl Matrigel and cultured in ENR-VC media with Valproic Acid ("V"; 1.5 mM, Sigma) and CHIR99021 ("C"; 3 μM, Selleckchem), containing Rho-kinase/ROCK inhibitor Y-27632 (10 μM) and Jagged1 (1 μM) only for the first 48 h. Under these culture conditions ISCs divide symmetrically without differentiating, growing into homogeneous stem cell colonies. ISC colonies were passaged once per week as single cells following a 3-min incubation with TrypLE containing 2 kU/ml DNase1, and Y-27632. Along with medium changes, treatment wells received different concentrations of rmIFNγ.

For co-culture of intestinal organoids with T cells, CD5+ cells were isolated from splenocytes using magnetic Microbeads with the MACS system (Miltenyi Biotec) according to the manufacturer's instructions. T cell purity was determined by flow cytometry and was routinely approximately 90%. T cells were cultured at a concentration of $1 \times 10^5$ T cells per well with 5 μg/ml plate-bound anti-CD3 monoclonal antibodies (mAbs, BD Pharmingen cat. #553058) and 2 μg/ml anti-CD28 mAbs (BD Pharmingen cat. #557393). After 3-5 days of culture, T cells were then harvested and cultured with passaged single organoid cells in Matrigel at a ratio of 0.5-50:1 T cells-to-organoid cells, Anti-IFNγ neutralizing antibodies were added to the culture at 10 μg/ml (Bio X Cell cat. #BP0055).

Human healthy duodenal organoids were cultured from banked frozen organoids (> passage 7) that had been previously generated from biopsies obtained during duodenoscopy of healthy human controls. All healthy controls had been investigated for celiac disease but turned out to have normal/non-pathologic histology. They had previously provided written informed consent to participate in such research according to a protocol reviewed and approved by the ethical review board (METC) of the UMC Utrecht, the Netherlands (STEM study: METC 10-402/K, and Metabolic Biobank: METC 19-489).

Human organoids were passaged via single cell dissociation using 1× TrypLE express (Gibco, Life Technologies). Single cells were resuspended in medium without growth factors (GF-) comprised of Advanced DMEM/F12 (GIBCO), 100 U/ml penicillin-streptomycin (GIBCO), 10 mM HEPES (GIBCO) and Glutamax (GIBCO) together with 50-66% Matrigel (BD Biosciences) and plated on pre-warmed 24- or 48-well cell culture plates (Costar). After Matrigel polymerization, organoid culture medium (hSI EM) was added consisting of GF- medium, Wnt-conditioned medium (25 or 50% final concentration), R-spondin-conditioned medium (20% final concentration), Noggin-conditioned medium (10% final concentration), 50 ng/ml murine EGF (Peprotech), 10 mM nicotinamide (SIGMA), 1.25 mM N-acetyl (Sigma), B27 (Gibco), 500 nM TGF-β inhibitor A83-01 (Tocris), and 10 uM P38 inhibitor SB202190 (Sigma). For single cells, 10 mM ROCK inhibitor Y-27632 (Abcam) was added for the first 2–3 days of the culture. Media was refreshed every 2–3 days. Along with media changes, treatment wells received different concentrations of rhIFNγ (R&D systems).

For human co-cultures with T cells: 500 single cells from TrypLE-dissociated organoids were cultured with activated human T cells in a ratio of 1:1 (= 500 T cells, 0.5 in the figures) and 1:50 (= 25,000 T cells, 25 in the figures) where applicable. Single cells were added to T cells and plated together in 50% Matrigel on pre-warmed 24- or 48-well cell culture plates. Media containing human Interleukin-2 (Proleukin; 12 IE/ml, Prometheus) and, where applicable, anti-IFNγ neutralizing antibodies (50 μg/ml, eBioscience cat. #16-7318-85), were added to the co-cultures.

T cells were isolated from human blood that was collected from healthy donors at the UMC Utrecht as approved by the UMC Utrecht's Ethics Committee under protocol number 07/125. After Ficoll gradient separation, CD4+ T cells were isolated from the peripheral blood mononuclear cells (PBMCs) using the MagniSort human CD4 T cell enrichment kit (Thermo Fisher) with the BD IMag™ Cell Separation Magnet (BD Biosciences) according to the manufacturer's protocol. Subsequently, T cells were activated at a concentration of 1 million cells/ml with plate-bound anti-CD3 (0.8 μg/ml, Biolegend cat. #300414) and soluble anti-CD28 (0.8 μg/ml, BioLegend cat. #302914) antibody stimulation for 2–3 days. After seven days of mouse and human co-cultures, total organoid numbers per well were counted by light microscopy to evaluate growth efficiency. All human organoid experiments were performed with material from three independent donors except for the experiments shown in Fig. 3b, which used two independent donors.

R-spondin-1-transfected HEK293T cells were provided by C. Kuo. Cell lines were tested for mycoplasma and confirmed to be negative. To image organoids and ISC colonies, random representative non-overlapping images of organoids and colonies were acquired from each well using a Zeiss Axio Observer Z1 inverted microscope or LSM880 (Carl Zeiss) with a 20x/0.8NA objective. For size evaluation, the images were analyzed using ImageJ software version 1.52.

### Immunohistochemical staining

Formalin-fixed tissue sections were deparaffinized with SafeClear II (Fisher HealthCare), antigen retrieval was performed with sodium citrate buffer (Ventana Medical Systems) and sections were stained using VECTASTAIN Elite ABC HRP Kit (Vector Laboratories) according to the manufacturer's instructions. Slides were incubated with anti-Ki67 (Cell Signaling, 1 μg/ml cat. #12202S) antibodies overnight, followed by a 60-minute incubation with biotinylated goat anti-rabbit IgG (Vector Laboratories cat. #PK-6101) at 1:200 dilution. Detection was performed with ImmPACT® AMEC Red Substrate, Peroxidase (HRP) (Vector Laboratories) according to the manufacturer's instructions. Slides were counterstained with Hematoxylin QS Counterstain

(Vector Laboratories), and coverslips were added with VectaMount® AQ Aqueous Mounting Medium (Vector Laboratories).

## Immunofluorescent staining

Immunofluorescent staining was performed at the Molecular Cytology Core Facility of MSKCC using a Discovery XT processor (Ventana Medical Systems). Formalin-fixed tissue sections were deparaffinized with EZPrep buffer (Ventana Medical Systems), antigen retrieval was performed with CC1 buffer (Ventana Medical Systems), and sections were blocked for 30 min with Background Buster solution (Innovex). After blocking with Background Buster solution (Innovex), the sections were followed by avidin-biotin blocking for 8 min (Ventana Medical Systems). To perform staining, sections were incubated with anti-c-Myc (Abcam, 1 μg/ml cat. #ab32072) for 5 h, followed by a 60 min incubation with biotinylated goat anti-rabbit IgG (Vector labs cat. #PK-6101) at 1:200 dilution. The detection was performed with Streptavidin-HRP D (part of DABMap kit, Ventana Medical Systems), followed by incubation with Tyramide Alexa-488 (Invitrogen, B40953) prepared according to the manufacturer instructions with pre-determined dilutions. After staining, slides were counterstained with DAPI (Sigma Aldrich, 5 μg/ml) for 10 min and coverslipped with Mowiol.

## 3-D imaging

Mouse small intestines were fixed for 3-D imaging by paraformaldehyde (4%) perfusion. The fixed tissues were immersed in 2% Triton-X 100 solution for permeabilization. Prior to staining, tissues were blocked with the blocking solution. Small intestine tissues were incubated with primary antibodies at 1:100 dilution. The primary antibodies used were anti-Olfm4 (Cell Signaling Technology, cat# 39141) and anti-c-Myc (R&D Systems, cat# AF3696). Alexa-Fluor-647-conjugated donkey anti-rabbit antibodies and Alexa-Fluor-546-conjugated donkey anti-goat antibodies (1:250, Invitrogen cat# A-31573 and A-11056) were used as secondary antibodies to reveal the immunopositive structure. Tissues were then incubated with DAPI (20 μg/ml, Invitrogen) to label nuclei. Finally, labeled specimens were immersed in FocusClear solution (CelExplorer, Hsinchu, Taiwan) for optical clearing before being imaged via confocal microscopy (Zeiss LSM 880). Amira 6.0.1 image reconstruction software (FEI) was used for 3-D processing and projection of the confocal images.

## Flow cytometry

For flow cytometry of small intestine organoid cells, organoids were dissociated using TrypLE (37 °C). After vigorously pipetting through a p200 pipette causing mechanical disruption, the crypt suspension was washed with 10 ml of DMEM/F12 medium containing 10% FBS and 2 kU/ml DNase1 and passaged through a 40 mm cell strainer.

DAPI and anti-Ki67 (1:50, BD Pharmingen cat# 561277) antibodies were used for intracellular staining for cell cycle analysis after fixation and permeabilization with a Fixation/Permeabilization kit (eBioscience) according to the manufacturer's instructions. See Supplementary Table 1 for the full list of antibodies used in this study. Flow cytometry analyses were performed with an LSR-II or X-50 cytometer (BD Biosciences) using FACSDiva version 6.1.2. (BD Biosciences), and the data were analyzed with FlowJo software 9.9.4 (Treestar).

## Quantitative PCR

For quantitative (q)PCR, RNA was isolated from organoids after ex vivo culture or crypts isolated from BMT recipients. Extracted RNA was also stored at −80°C. Reverse transcriptase PCR (RT–PCR) was performed with a QuantiTect Reverse Transcription Kit (QIAGEN) or a High-Capacity RNA-to-cDNA Kit (Applied Biosystems) for mouse, and iScript cDNA Synthesis Kit (BioRad) for human samples.

For mouse genes, specific primers were obtained from Applied Biosystems: *Gapdh*: Mm99999915_g1; *Myc*: Mm00487804_m1; *Axin2*: Mm00443610_m1; *Irf1*: Mm01288580_m1; *Ccnd1*: Mm00432359_m1. Some primers were obtained from PrimerBank: *Gapdh* (ID 6679937a1), *Ccnd1* (ID 6680868a1). Other primers were obtained from ITD using published sequences or after primer-design with Primer3 as follows; *Paf*[39]: *Fw TGTGATCAGGTTGCAAAGGA and Rv TTCAGGCTGTCCCCT AA AGA, Irf1_P1: Fw GGGAATCCCGCTAAGTGTTT* and *Rv CTACCTCGACGA AGGAGTGG, Myc_P1: Fw AGGGAGACCTACAGGGGAAA and Rv CACACAC ACTCCAGCACCTC, Myc_P2: Fw AGCGAGAGACAGAGGGAGTG* and *Rv TCCAGAGCTGCCTTCTTAGG, Myc_P3: Fw TCCAGGGTACATGGCGTATT* and *Rv TCGGCTGAACTGTGTTCTTG, and* negative control: *Fw TAGCC AGAAGCTGGAAAGAAGCCA and Rv TGATACCCTCCAGGTCCAACCATT.*

For human genes, specific primers were obtained from Integrated DNA Technologies after primer-design with the NCBI nucleotide and primer blast databases and having checked them for effectiveness: *HP1BP3: Fw CCCACGTCCCAAGATGGAT and Rv CTGATGCACCACTCT TCTGAA, MYC: Fw CCTACCCTCTCAACGACAGC and Rv CTTGTT CCTCCTCAGAGTCGC, CCND1: Fw ATCAAGTGTGACCCGGACTG and Rv CTTGGGGTCCATGTTCTGCT.*

cDNAs were amplified for mouse primers with TaqMan or SYBR master mix (Applied Biosystems) in QuantStudio 7 Flex System (Applied Biosystems) and for human samples with a SYBR master mix (BioRad) in a CFX96™ Real-Time PCR Detection System (BioRad). Relative amounts of mRNA were calculated by the comparative ΔCt method with *Gapdh* as house-keeping gene for mouse samples and with *HP1BP3* for human samples.

## Chromatin immunoprecipitation

ISC colonies cultured for 7 days were treated with 1 ng/mL of IFNγ for 3 h. DNA and proteins were cross-linked for 8 min using 1% formaldehyde. Chromatin immunoprecipitation (ChIP) was performed as previously described[50,51], using rabbit anti-STAT1 antibodies (Cell Signaling, cat# 9172) or rabbit IgG isotype antibodies (Cell Signaling, cat# 2729) at a dilution of 1:50. ChIP was performed on 95% of each sample, with the remaining 5% used as input. qPCR was performed on input and ChIP samples for indicated targets. The qPCR threshold cycle (CT) values from input samples were adjusted to 100% (CT-4.332), after which the % of input was calculated as $100 \times 2^{(Adjusted\ Input\ CT\ -\ ChIP\ CT)}$. The difference (Δ) in the % of input of anti-STAT1 ChIP over isotype ChIP was then computed to determine enrichment of qPCR targets. Enrichment was controlled by qPCR and analysis of *Foxp3* in input and ChIP samples.

## IFNγ treatment in vivo

RiboTag experiments evaluating gene expression in Olfm4+ ISCs were performed as previously described[44]. Olfm4-Ribo mice were administered IFNγ (20 μg) or vehicle (PBS) via intraperitoneal injection twice, 24 and 48 h prior to tissue harvest. In addition, Olfm4-Ribo mice were administered tamoxifen (2 mg/mouse) to induce hemagglutinin labeling of ribosomes. The tamoxifen injection was performed 20 h prior to tissue harvest in order to restrict Cre-induced hemagglutinin expression to Olfm4+ ISCs[44]. The IFNγ dosing strategy was determined by reducing the amount of IFNγ that was administered in comparison to what has previously been shown to cause intestinal toxicity[24]. Based on mouse availability, Olfm4-Ribo mice were 2-9 months old at the time of analysis; IFNγ-treated mice were age-matched to PBS-treated controls for determining relative gene expression. Following harvest of the distal 10 cm of small intestine, and thus enriching for terminal ileum, the tissue was homogenized as previously described[44], and hemagglutinin-labeled ribosomes were isolated using anti-hemagglutinin-labeled magnetic beads (Pierce Thermo Scientific; cat #88836). RNA was then purified from the isolated ribosomes for qPCR analysis of transcripts from Olfm4+ ISCs[44].

## Statistics

No statistical methods were used to predetermine sample size. To detect an effect size of >50% difference in means, with an assumed coefficient of variation of 30%, common in biological systems, we attempted to have at least five samples per group, particularly for in vivo studies. All experiments were repeated at least once, unless otherwise stated. No mice were excluded from experiments.

Graphs indicate the mean and standard error of the mean (S.E.M.) for the various groups. Statistics are based on "n" biological replicates. All statistical tests performed were two-sided. For the comparisons of two groups, a t test or non-parametric test was performed. All analyses of statistical significance were calculated and displayed in comparison with the reference control group unless otherwise stated.

Statistical analyses of organoid numbers were based on individual wells. To account for intra-individual and intra-experimental variation as well, all ex vivo organoid experiments were performed at least twice with several wells per condition, and with sample material coming from at least two different mice or two different human donors. Statistics were calculated and display graphs were generated using Graphpad Prism version 10. P <.05 was considered statistically significant.

## Reporting summary

Further information on research design is available in the Nature Portfolio Reporting Summary linked to this article.

## Data availability

Graphical data generated in this study, including data presented as bar graphs, line graphs, or overlapping data points, have been included as raw data in the Source Data file. The single cell RNA sequencing data generated for this study have been deposited in the Gene Expression Omnibus under the accession number GSE276126. This study also utilized the publicly available dataset GSE83394. Source data are provided with this paper.

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

## Acknowledgements

We thank Karuna Ganesh, Dana Pe'er, and Herman Gudjonson for their valuable assistance. We gratefully acknowledge the support of the Memorial Sloan Kettering Cancer Center (MSKCC) Single Cell Research Initiative, Research Animal Resource Center, Molecular Cytology Core Facility, and Integrated Genomics Operation Core, with funding from the MSKCC NCI Cancer Center Support Grant (P30 CA08748), Cycle for Survival, and the Marie-Josée and Henry R. Kravis Center for Molecular Oncology. This research was supported by National Institutes of Health award numbers R01-HL125571, R01-HL146338, and R01-HL145631 (A.M.H), and MSKCC Core Grant P30-CA008748. Support was also received from the Susan and Peter Solomon Divisional Genomics Program, the Ludwig Center for Cancer Immunotherapy, the Parker Institute for Cancer Immunotherapy, the Anna Fuller Fund, and the MSK Sawiris Foundation (A.M.H.). S.T. was supported by a DKMS John Hansen Research grant, Y.F. was supported by the Amy Strelzer Manasevit Research Program, V.A. was supported by the German Research Foundation (DFG), S.A.J. was supported by the Alexandre Suerman Stipend of the UMC Utrecht, and C.A.L. was supported by the WKZ fund of the UMC Utrecht.

## Author contributions

S.T. designed, performed, and analyzed in vivo and ex vivo experiments and drafted the manuscript. R.S. performed bioinformatic analysis and drafted the manuscript. W.C., M.C., and Y.F. led additional in vivo and ex vivo experiments. S.A.J. performed and analyzed human ex vivo experiments. T.I. performed and analyzed in vivo experiments. E.S. and J.Sa. assisted with ChIP experiments. A.E. and J.K. performed and monitored bone marrow transplants and maintained the mouse colonies. V.A. provided input and helped additional experiments. O.C. assisted with scRNA-seq experiments. H.G. performed bioinformatic analysis. C.L. analyzed intestinal histopathology. H.I., J.Su., N.R., L.M., C.A.L., and A.M.H. supervised the research.

## Competing interests

The authors declare no competing financial interests. A.M.H. and C.A.L hold intellectual property related to Interleukin-22, and A.M.H. has a collaboration with Evive Biotechnology (Shanghai) Ltd, which supported a multicenter clinical trial studying use of Interleukin-22 in patients with GVHD. A.M.H. also serves in a volunteer capacity as a member of the Board of Directors of the American Society for Transplantation and Cellular Therapy (ASTCT).

## Additional information

[1]Department of Medicine and Human Oncology & Pathogenesis Program, Memorial Sloan Kettering Cancer Center, New York, NY 10065, USA. [2]Department of Hematology, NHO Kyushu Medical Center, Fukuoka, Fukuoka 810-8563, Japan. [3]New York Genome Center, New York, NY 10013, USA. [4]Immunology & Microbial Pathogenesis Program, Memorial Sloan Kettering Cancer Center, New York, NY 10065, USA. [5]Division of Pediatrics, Regenerative Medicine Center, University Medical Center Utrecht, Utrecht University, 3508 AB Utrecht, The Netherlands. [6]Princess Máxima Center for Pediatric Oncology, 3584 CS Utrecht, The Netherlands. [7]Computational and Systems Biology, Memorial Sloan Kettering Cancer Center, New York, NY 10065, USA. [8]Department of Pathology, Yale School of Medicine, New Haven, CT 06520, USA. [9]Institute of Biotechnology Vilnius University, Vilnius LT-10257, Lithuania. [10]Department of Medicine, Weill Cornell Medical College, New York, NY 10065, USA. [11]These authors contributed equally: Shuichiro Takashima, Roshan Sharma, Winston Chang, Marco Calafiore, Ya-Yuan Fu. ✉e-mail: hanasha@mskcc.org

