## [Peer Review File · Nature Communications]

REVIEWER COMMENTS

Reviewer #1 (Remarks to the Author):

The authors have appropriately addressed my points raised in the previous review. The paper is ready for publication.

Reviewer #2 (Remarks to the Author):

The authors performed additional experiments, in which epithelial IFN γ R-deficient or IFNAR-deficient mice were transplanted from allogeneic donors. Unlike in epithelial IFN-gR-deficient mice, where allogeneic BMT did not boost gut epithelial cell proliferation, epithelial cells in IFNAR significantly increased in proliferation following allogeneic BMT (new Figure 2i-k). Combined with the RNA-seq data (new Figure S1f), where IFN-g-specific genes, but not IFN-a-specific genes, were predominantly upregulated in allogeneic recipients. These findings confirm the pivotal role of IFN-g, rather than IFN-a, in the enhanced proliferation of epithelial cells after allogeneic BMT.

Additional experiment, where Olfm4-Cre x RiboTag mice were injected with a low dose of IFN γ , demonstrated that ISC upregulated both Myc and CyclinD1 without causing major tissue injury. This finding rules out a significant role for IFN γ -induced tissue injury in the enhanced proliferation induced by IFN γ (new Figure S6).

IFN γ may either skew the differentiation of ISCs or induce the proliferation of other epithelial cells. However, this reviewer agrees that such topics are beyond the scope of this paper.

In new Figure 5e, the authors optimized IFN-g concentration and found that a higher concentration of IFN-g induces CCND1 expression, as shown in mouse organoid experiments. The authors believe that a higher IFN-g concentration was required to overcome Wnt-induced CCND1 in human organoids due to the Wnt-rich culture conditions for human organoids.

MYC was not detected in ISCs at the crypt base in the immunofluorescent study (Figure 6e). The authors improved imaging by using 3-D whole-mount microscopy and found that MYC was induced in Olf4+ ISCs at the crypt base in GVHD mice compared to control mice without GVHD (new figure 4ef).

Overall, my concerns were thoroughly addressed, and the additional experiments provided mostly satisfactory responses to my queries. This reviewer has only one concern as following.

Major comment;

1. It is convincing that STAT1 deficiency mitigated ISC damage, likely caused by a higher concentration of IFN γ early after allo-BMT, while STAT1 deficiency also suppressed the proliferation of ISCs maintained by a lower concentration of IFN γ later point. This culminated in the reduction of ISCs on day 14. However, it is still not clear why these STAT1-deficient mice died after allo-BMT. If the impaired mucosal regeneration is the cause of death, the authors should test whether membrane integrity is worse in STAT1-deficient recipients compared to WT recipients by using the FITC-dextran assay, or etc.

REVIEWER COMMENTS

Reviewer #1 (Remarks to the Author):

The authors have appropriately addressed my points raised in the previous review. The paper is ready for publication.

Thank you. We appreciate the thoughtful evaluation that has improved the quality of this work.

Reviewer #2 (Remarks to the Author):

The authors performed additional experiments, in which epithelial IFN γ R-deficient or IFNAR-deficient mice were transplanted from allogeneic donors. Unlike in epithelial IFN γ R-deficient mice, where allogeneic BMT did not boost gut epithelial cell proliferation, epithelial cells in IFNAR significantly increased in proliferation following allogeneic BMT (new Figure 2i-k). Combined with the RNA-seq data (new Figure S1f), where IFN-g-specific genes, but not IFN-a-specific genes, were predominantly upregulated in allogeneic recipients. These findings confirm the pivotal role of IFN-g, rather than IFN-a, in the enhanced proliferation of epithelial cells after allogeneic BMT.

We agree, and we appreciate the encouragement to pursue this important experiment.

Additional experiment, where Olfm4-Cre x RiboTag mice were injected with a low dose of IFN γ , demonstrated that ISC upregulated both Myc and CyclinD1 without causing major tissue injury. This finding rules out a significant role for IFN γ -induced tissue injury in the enhanced proliferation induced by IFN γ (new Figure S6).

Thank you. In the prior submission, based on mouse availability, these data included 3-4 Olfm4-Cre x RiboTag (Olfm4-Ribo) mice per group combined from two experiments. Since then, additional Olfm4-Ribo mice have become available to repeat this experiment for a third time, treating Olfm4-Ribo mice with either IFN γ or PBS and evaluating ISC gene expression for *Ccnd1* and *Myc*. The results were consistent with our previous findings, demonstrating that IFN γ treatment was sufficient to increase c-Myc expression in ISCs and induce gene expression consistent with increased ISC proliferation. We have now combined the data from all three experiments together, and with the increased n's further supporting these findings, we have now moved the data into the main figures in a **new Figure 6f**.

IFN γ may either skew the differentiation of ISCs or induce the proliferation of other epithelial cells. However, this reviewer agrees that such topics are beyond the scope of this paper.

In new Figure 5e, the authors optimized IFN-g concentration and found that a higher concentration of IFN-g induces CCND1 expression, as shown in mouse organoid experiments. The authors believe that a higher IFN-g concentration was required to

overcome Wnt-induced CCND1 in human organoids due to the Wnt-rich culture conditions for human organoids.

Thank you. It is indeed possible that a higher concentration is necessary to identify IFN γ -induced *CCND1* upregulation. We note however that the kinetics of *Myc* upregulation and *Ccnd1* upregulation appear to be slightly different, with *Myc* levels increasing prior to *Ccnd1*. As the main focus of this experiment was to look for changes in *MYC* expression in response to IFN γ , it remains possible that IFN γ -dependent *CCND1* expression may also be identified at lower concentrations when assessed at other timepoints.

MYC was not detected in ISCs at the crypt base in the immunofluorescent study (Figure 6e). The authors improved imaging by using 3-D whole-mount microscopy and found that MYC was induced in Olf4+ ISCs at the crypt base in GVHD mice compared to control mice without GVHD (new figure 4ef).

We have found that MYC antibody staining can be challenging, and the increased resolution of our 3-D imaging approach was indeed helpful for evaluating MYC expression post-BMT.

Overall, my concerns were thoroughly addressed, and the additional experiments provided mostly satisfactory responses to my queries. This reviewer has only one concern as following.

Major comment;

1. It is convincing that STAT1 deficiency mitigated ISC damage, likely caused by a higher concentration of IFN γ early after allo-BMT, while STAT1 deficiency also suppressed the proliferation of ISCs maintained by a lower concentration of IFN γ later point. This culminated in the reduction of ISCs on day 14. However, it is still not clear why these STAT1-deficient mice died after allo-BMT. If the impaired mucosal regeneration is the cause of death, the authors should test whether membrane integrity is worse in STAT1-deficient recipients compared to WT recipients by using the FITC-dextran assay, or etc.

Thank you for the positive assessment of the revisions. We agree with the reviewer that there is a shift in the impact of epithelial IFN γ R deficiency over time post-BMT, particularly in the MHC-mismatched BMT model, which is associated with high levels of IFN γ early post-transplant. We also agree with the reviewer that while the epithelial STAT1-deficient mice ultimately do worse after allo-BMT, and they show a persistent reduction in epithelial proliferation and a reduction in ISC frequencies, it is not clear why they demonstrated more rapid mortality or if this is directly related to the reduction in epithelial proliferation. We have included a statement in the discussion highlighting this.

In addition, in response to the reviewer's suggestion to test the integrity of the intestinal barrier, we performed several new experiments in an attempt to examine this in detail. Although we did not have more epithelial STAT1-deficient mice available to be able to perform new BMTs and the FITC-dextran assay, we were able to examine material from the transplants we had already performed to look for evidence of bacterial translocation and neutrophilic infiltration within the intestinal mucosa, as measures of impaired barrier function. First, we performed qPCR for bacterial 16S ribosomal RNA on crypts isolated from STAT1-intact (STAT1-floxed, Cre-negative) and from epithelial STAT1-deficient (STAT1-floxed, Villin-Cre-positive) mice on day 21 after B10.BR-into-B6 allo-BMT (shown to the right). In both STAT1-intact and

STAT1-deficient mice, in qPCR data combined from two independent experiments, we could not detect the presence of bacteria within crypt epithelium isolated from the majority of mice. While there was a slight trend toward increased detection of bacterial 16S rRNA in STAT1-deficient BMT recipients, suggesting the possibility of reduced barrier function in these mice, the trend was modest and not statistically significant.

To take an alternative approach, we examined intestinal tissue histologically for neutrophilic infiltration within the intestines, which has been associated with bacterial translocation post-BMT (Schwab et al. *Nature Medicine* 2014). Small intestine tissue sections from transplanted mice were scored semi-quantitatively for neutrophil infiltration by a blinded pathologist. Once again, as shown to the right we found no statistically significant differences in neutrophilic infiltration within the small intestines between STAT1-intact and STAT1-deficient mice post-BMT.

Finally, as shown below, we took a third approach of directly examining bacterial invasion within the mucosa by performing fluorescent in situ hybridization (FISH) on intestinal sections using the EUB338 bacterial probe. While we were able to detect rare events of bacterial translocation within the mucosa using this probe, we once again found no difference in measurements of intestinal infiltration of EUB338⁺ bacteria within the intestines of STAT1-intact and epithelial STAT1-deficient mice.

Therefore, this series of experiments provides no clear evidence of worsened epithelial barrier function in the intestines of STAT1-deficient mice after B10.BR into B6 allo-BMT. It is possible that the complex combination of phenotypes induced by the IFN γ /STAT1 axis post-BMT, including both direct induction of ISC apoptosis and tissue damage early on as well as direct promotion of epithelial proliferation may be too noisy to detect effects on barrier function related to IFN γ -driven epithelial proliferation. We have therefore added a statement to the discussion indicating that the effects of IFN γ /STAT1 signaling on epithelial barrier function in the GI tract remain to be defined and must be pursued in future work. We thank you again for the thoughtful evaluation that has improved the quality of this study.